# A combined mathematical and experimental approach reveals the drivers of time-of-day drug sensitivity in human cells

Nica Gutu[1,2], Hitoshi Ishikuma [1], Carolin Ector[1,2], Ulrich Keilholz[1], Hanspeter Herzel[2,3] & Adrián E. Granada [1,4] ✉

The circadian clock plays a pivotal role in regulating various aspects of cancer, influencing tumor growth and treatment responses. There are significant changes in drug efficacy and adverse effects when drugs are administered at different times of the day, underscoring the importance of considering the time of day in treatments. Despite these well-established findings, chronotherapy approaches in drug treatment have yet to fully integrate into clinical practice, largely due to the stringent clinical requirements for proving efficacy and safety, alongside the need for deeper mechanistic insights. In this study, we employ a combined mathematical and experimental approach to systematically investigate the factors influencing time-of-day drug sensitivity in human cells. Here we show how circadian and drug properties independently shape time-of-day profiles, providing valuable insights into the temporal dynamics of treatment responses. Understanding how drug efficacy fluctuates throughout the day holds immense potential for the development of personalized treatment strategies aligned with an individual's internal biological clock, revolutionizing cancer treatment by maximizing therapeutic benefits. Moreover, our framework offers a promising avenue for refining future drug screening efforts, paving the way for more effective and targeted therapies across diverse tissue types.

Many forms of life have developed an internal timing process to adapt their activities to the environmental day-night cycles of 24-h periodicity. This biological rhythm is termed the circadian clock and mediates many essential physiological processes of the organism[1]. In mammals, it orchestrates the expression of up to 60% of protein-coding genes in various cell types, generating daily rhythms in main functions such as metabolism, immune response, DNA damage repair and proliferation[2–4]. Moreover, disruption of the circadian rhythms in humans negatively impact many physiological functions leading to several disorders such as insomnia, depression, metabolic diseases, and cancer[3–6].

Integrating a patient's circadian rhythm into cancer treatment strategies holds the promise of revolutionizing cancer care by minimizing toxicity and maximizing anti-tumor effects. Previous studies in mice have shown that circadian clock significantly modulates drug effectiveness throughout the day[7–28] and several circadian-modulation mechanisms have been proposed, such as circadian regulation of immune response[29–31], pharmacokinetics[27,32] and circadian modulation of stress response pathways[33,34]. While these studies have been invaluable for pinpointing the role of the circadian clock in time-of-day drug responses, their intricate nature constrains the capacity to isolate various confounding factors, limiting precise mechanistic understanding. As an alternative, in-vitro approaches have allowed to systematically characterize the time-of-day drug sensitivity profiles in a panel of well-defined tumor subtypes cell models. These investigations have shown that in most cases, the optimal time for drug administration depends on the specific tumor-subtype model and drug[24,35]. They highlight the complexity of circadian influences on drug efficacy and underscore the need for further research to develop tailored chronotherapy strategies adapted to specific tumor subtypes and treatments[24,36,37].

[1]Comprehensive Cancer Center, Charité - Universitätsmedizin Berlin, Charitéplatz 1, 10117 Berlin, Germany. [2]Humboldt Universität zu Berlin, Berlin, Germany. [3]Institute for Theoretical Biology, Berlin, Germany. [4]German Cancer Consortium (DKTK), Berlin, Germany. ✉e-mail: adrian.granada@charite.de

**Article**

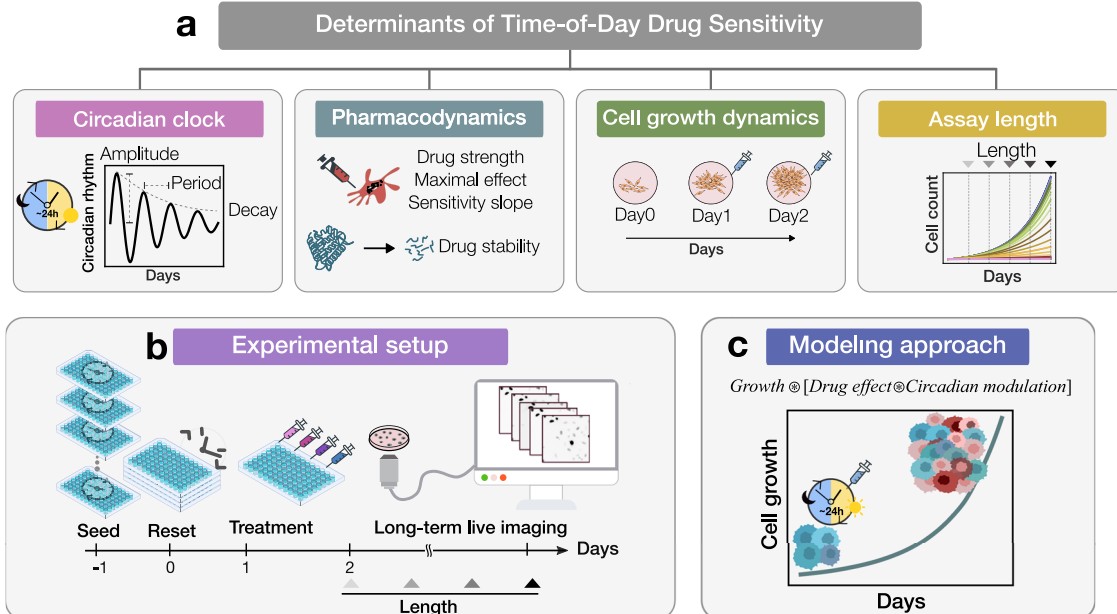

**Fig. 1 | Framework to study factors determining time-of-day drug sensitivity.**
**a** Factors determining time-of-day drug sensitivity, which can be biological and
empirical. The biological factors comprise the circadian clock, pharmacodynamics,
and cell growth dynamics. The empirical factors are cell growth dynamics and the
assay length. **b** Experimental setup as part of the framework to determine the time-
dependent factors driving the time-of-day drug responses. **c** Computational
approach as part of the framework to study the factors shaping time-of-day drug
sensitivity. The model considers population growth under pharmacological treat-
ment with a circadian clock modulation.

Both circadian and non-circadian factors intricately weave together to
shape time-of-day drug responses. The circadian clock exerts influence over
various cellular processes, such as drug metabolism, cell cycle regulation,
DNA repair mechanisms, and drug transport. It orchestrates the expression
of enzymes like cytochrome P450, leading to daily fluctuations in enzyme
levels that profoundly impact drug metabolism rates[27,28,38–42]. This rhythmic
enzymatic activity can accelerate drug clearance during peak times,
potentially reducing its efficacy. Moreover, the circadian clock synchronizes
cell cycle phases[43–45], a critical factor in determining the effectiveness of
chemotherapeutic agents, as the timing of drug administration can sig-
nificantly influence cell sensitivity[46,47]. Circadian rhythms also modulate
DNA repair mechanisms[48–51], which regulation can either enhance or
diminish a drug's efficacy by influencing cellular resistance to DNA-
damaging treatments. Furthermore, the circadian control over drug trans-
port proteins affects intracellular drug concentrations, thereby influencing
drug potency.

In addition to circadian influences, other non-circadian cellular pro-
cesses play pivotal roles in shaping time-of-day drug responses. Pharma-
cokinetic properties such as a drug's half-life, bioavailability, and clearance
rate interact with circadian rhythms, impacting how drugs are absorbed,
metabolized, and excreted over the course of a day. The proliferation rate of
tumor cell models also significantly impacts drug sensitivity; rapidly
dividing cells may exhibit different response patterns compared to slower-
growing or quiescent cells. Moreover, the duration of experimental assay to
estimate time-of-day responses is critical, ranging from short-term studies
of up to 48 h to longer investigations spanning multiple circadian cycles.
Such durations are essential to comprehensively capture both immediate
and prolonged drug effects, observing daily rhythms and extended impacts
on cellular responses.

This paper explores the influence of circadian and non-circadian cel-
lular processes on drug sensitivity over the course of a day, with a specific
focus on in-vitro characterization. Distinct factors including characteristics
of the circadian clock signal such as amplitude, period, and amplitude decay
rate, pharmacodynamic properties of specific drugs, growth attributes of the
cell-line model, and the duration of assays are examined for their roles in

time-of-day (ToD) drug response profiles (Fig. 1a). To this end, we have
implemented an experimental framework that employs live cell imaging and
computational image analysis, specifically tailored to evaluate time-
dependent drug responses in vitro (Fig. 1b). Our study further introduces
a generic modeling approach that simulates the oscillatory influence of
circadian modulation on effective drug concentration, incorporating both
exogenous and endogenous factors (Fig. 1c). This model helps to clarify the
impacts of non-circadian factors such as cell growth and drug response
characteristics on ToD response curves, offering new insights that advance
our understanding of chronotherapy. By computationally exploring het-
erogeneous cell populations, we identify critical factors that influence
optimal treatment timing, thereby enhancing the efficacy of time-specific
drug interventions. These contributions significantly advance the
mechanistic understanding of timing in drug screening and provide a robust
foundation for tailoring drug administration to circadian rhythms.

## Results
### How circadian clock properties shape time-of-day drug sensitivity

The body's internal clock modulates various processes that influence the
effects of drugs, including their transport and metabolism, cell growth, and
DNA repair mechanisms. These factors influence the overall effectiveness of
drugs, essentially controlling their strength. To study these effects in a
unified approach, we developed a mathematical model where the circadian
clock acts as a modulator of the effective drug concentration, by boosting or
attenuating a baseline drug dose—referred to here as the reference con-
centration—at different times of the day (see Fig. 2a and Methods). Using
this approach, we compared cell growth at a standard drug concentration to
growth at boosted and attenuated concentrations (Fig. 2b left). Our model
describes variations in drug effects at different times of the day based on
simulated concentrations, though actual time-of-day effects in biological
organisms have not been directly tested in this work. Using this model, the
growth curves show cells growing at slightly different rates depending on the
time of day (Fig. 2b right). Calculating relative growth throughout the day
yields time-of-day (ToD) response plots, highlighting sensitive and resistant

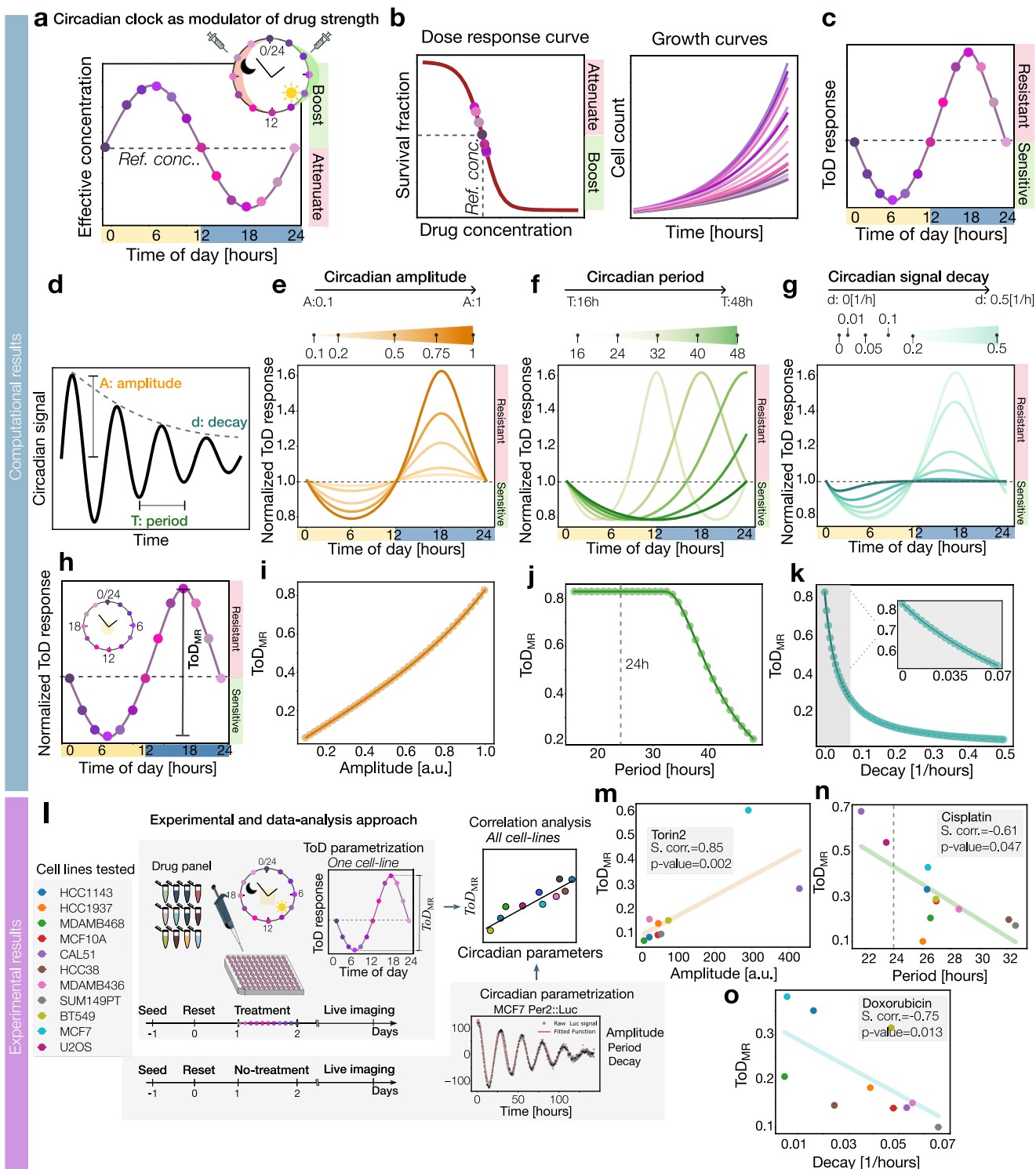

phases (Fig. 2c). Due to the exponential nature of cell growth, linear differences in effective drug concentration led to exponential growth differences, creating asymmetries in the ToD curves. In this study, we examine the effects of both cytostatic and cytotoxic drugs. For clarity, in the following section we studied the effect of a cytotoxic drug.

Building on this, our modeling framework abstracts the source of circadian modulation, treating it as an oscillatory modulation of the effective drug concentration. This abstraction allows the model to incorporate both exogenous and endogenous influences, such as circadian processes affecting drug delivery to tumor cells or endogenous circadian clocks influencing

drug uptake dynamics or stress response pathways. By focusing on the effective drug concentration as the key variable, this approach provides a unified and versatile framework to examine how oscillatory properties interact with drug characteristics to shape time-of-day-dependent outcomes. This avoids requiring explicit assumptions about the origin of the modulation while capturing its downstream effects.

Next, we investigated how various characteristics of the circadian clock—such as amplitude, period, and amplitude decay rate—modulate effective drug concentration and influence time-of-day (ToD) responses (Fig. 2d). We simulated the ToD response for increasing amplitude of circadian

**Fig. 2 | Circadian clock properties shape the time-of-day drug response.**
**a** Circadian clock modulation of drug administration at different hours within 24 h with the corresponding effective concentration. Color labeling indicates different administration hours. **b** Left: population survival fraction for different drug concentrations indicating the reference concentration. Right: cell growth sketch for drug administrated at different hours within a day. **c** Time-of-day (ToD) sketch for drug administration at different hours within 24 h using the same color coding as in (**a**). **d** Diagram of an exponentially decaying sinusoidal function for the circadian clock with the amplitude, period, and decay definitions. **e** Simulated ToD response curve changing the amplitude values of the circadian clock. The color gradient corresponds to an increasing range of values from 0.1 to 1. The responses above 1 (dashed line) are defined as resistant (red) and below are defined as sensitive (green). **f** Simulated ToD response curve for different circadian period values. **g** Simulated ToD response curve for different circadian signal decay values. **h** ToD sketch with the definition of the maximum range. **i** Maximum range of ToD response curve for different circadian amplitudes from (**e**). **j** Maximum range of the ToD response curve for different circadian periods from (**f**). **k** Maximum range of the ToD

response curve for different decay values from (**g**). Top right: zoom of the maximum range of the ToD within the gray range [0, 0.07]. **l** Experimental setup of ToD drug assays and bioluminescence recordings of different tumor cell lines with the posterior parametrization to obtain the circadian and ToD properties and perform the correlation analysis. **m** Maximum range of the experimental ToD response curve for the drug Torin2 versus the circadian amplitudes (of Bmal1 estimate) of the cell lines from (**l**). The legend inside the plot shows the Spearman correlation coefficient with the corresponding $p$-value. The Pearson correlation coefficient with the corresponding $p$-value is 0.72 and 0.02. **n** Maximum range of the experimental ToD response curve for Cisplatin versus the circadian period (of Per2 estimate) of the cell lines from (**l**). The legend inside the plot shows the Spearman correlation coefficient with the corresponding $p$-value. The Pearson correlation coefficient with the corresponding $p$-value is −0.74 and 0.01. **o** Maximum range of the experimental ToD response curve for Doxorubicin versus the circadian signal decay values (of Bmal1 estimate) of different cell lines (legend **l**). The legend inside the plot shows the Spearman correlation coefficient with the corresponding $p$-value. The Pearson correlation coefficient with the corresponding $p$-value is −0.64 and 0.045.

---

oscillations (Fig. 2e and Supplementary Fig. S1a). Although the circadian modulation of effective drug concentration is symmetric, as shown in Fig. 2a, its effect on cellular growth curves results in asymmetrical areas (Fig. 2e). This asymmetry arises because variations in drug levels around a reference concentration interact with the exponential nature of growth and the shape of the dose-response curve. Therefore, equal proportional increases and decreases in drug concentration led to disproportionate effects, creating uneven, asymmetrical regions of resistance and sensitivity on the time-of-day response curves.

Examining the role of circadian period revealed that longer periods shift and broaden circadian oscillations (Fig. 2f and Supplementary Fig. S2b). Finally, the decay timescale of the circadian signal influences the ToD curves, with fast-decaying signals producing diminished curves and slower decays leading to more uniform ToD drug responses throughout the day (Fig. 2g and Supplementary Fig. S1c).

Next, we calculated the maximum range of the Time-of-Day (ToD) drug response for each of these parameters (Fig. 2h). The maximum range value can be used as an estimate of the overall relevance of circadian-based approaches for a given drug cell line combination. This analysis shows that, as expected, the maximum range of the ToD response curve increases proportionally with the amplitude of the circadian signal (Fig. 2i). The analysis of the period indicates that up to approximately 32 h, the maximum range of the ToD response remains flat, followed by a gradual linear decrease beyond this duration (Fig. 2j). This simulations extends beyond the typical human circadian period of 22–26 h to explore wider ranges observed in certain in vitro tumor cell models (Supplementary Fig. S1d). This allows us to examine how unusually longer circadian periods might influence drug efficacy. Finally, we examine the role of the circadian amplitude decay rate—also known as amplitude dampening—which quantifies how quickly the circadian signal's strength diminishes. Analysis shows that this decay rate significantly influences the ToD response curve: we observe a linear decrease in the curve's maximum range at lower decay rates, which shifts to an exponential decline as the decay rate increases (Fig. 2k). It is important to note that our model assumes that drug effects, once determined by their time-of-day of application, remain constant over the course of treatment, not accounting for potential variations in drug activity or cellular responses that might occur due to evolving and changing circadian properties or prolonged drug exposure.

To study how these simulation results relate to experimentally evaluated ToD curves, we built upon and expanded the analysis of a recent set of experiments from our group, originally published in ref. 37. This work involved using real-time luciferase reporters of Per2 and Bmal1 to characterize the circadian clock in various tumor cell lines. From each recording we then assessed the circadian amplitude, period, and decay in each cell line. In the same study, we also evaluated the ToD responses of all these cell lines against a panel of eight chemotherapeutic drugs. Here, we analyzed these circadian parameters to explore their correlation with the ToD maximum

range of drug effectiveness for each chemotherapeutic agent, as illustrated in Fig. 2l. The results of these studies were detailed in Supplementary Table 1. The drugs with the highest correlation coefficients were Torin2, Cisplatin, Doxorubicin, and Alpelisib. In Fig. 2m, we show the analysis results for Torin2, the PI3K/AKT/mTOR inhibitor, which exhibits a positive correlation between the maximum range of the ToD response curve and amplitude values, consistent with our simulations (Fig. 2i). Cisplatin and Doxorubicin showed a negative correlation between the maximum range of the ToD response curve and period and decay values, respectively (Fig. 2n–o). Alpelisib followed a similar trend, displaying a positive correlation with amplitude values and negative correlations with period and decay values (Supplementary Fig. S1e). Hence, these experimental results suggest that circadian modulation of certain drugs aligns with a model where the circadian clock modulates effective drug concentration.

It is important to clarify that in this analysis we assessed the amplitude of a single circadian reporter as a proxy to evaluate the circadian clock's impact on drug, which is clearly a simplification of the complex interactions within the circadian system. Nonetheless, our findings provide an initial approximation of the circadian clock's influence and that allow us to explore the relationships between circadian parameters and ToD drug response parameters.

## How drug properties modulate the time-of-day responses
In addition to the circadian clock influencing drug efficacy, the properties of the drugs themselves can significantly shape the profile of time-of-day responses. To understand how specific drug characteristics impact these profiles, we employed our mathematical model to analyze the effects of variations in concentration values, maximal drug effect, sensitivity slope, and drug stability on time-of-day responses (Fig. 3a). We began by examining how drug concentrations affect time-of-day (ToD) plots for a cytotoxic drug. Typically, to study time-of-day effects, a constant drug dose—referred to here as the reference concentration—is administered at different times of the day, and their relative effects are compared throughout the day. In this study, we considered reference doses at various points along a dose-response curve (Fig. 3b). Our simulations indicated that the amplitude of the time-of-day response curve increases with rising dose until reaching a concentration near the half-maximal drug effect, as illustrated by doses I to III in Fig. 3c and Supplementary Fig. S2a. Beyond this point, the amplitude of the ToD response begins to decrease, from doses IV and V. Calculating the maximum range for the whole range of doses results in a quadratic relationship (Fig. 3d).

We next analyzed the role of the maximal drug effect on population growth ToD curves. For cytotoxic drugs, we examined the effect of reference doses at half-maximal drug effect from drugs with dose-response curves corresponding to a gradually decreasing fraction of surviving cells (killing rate, $k_l$), which correlates to a decrease in the highest drug concentration ($E_{max}$). Strong cytotoxic drugs with smaller $E_{max}$ values reduced survival

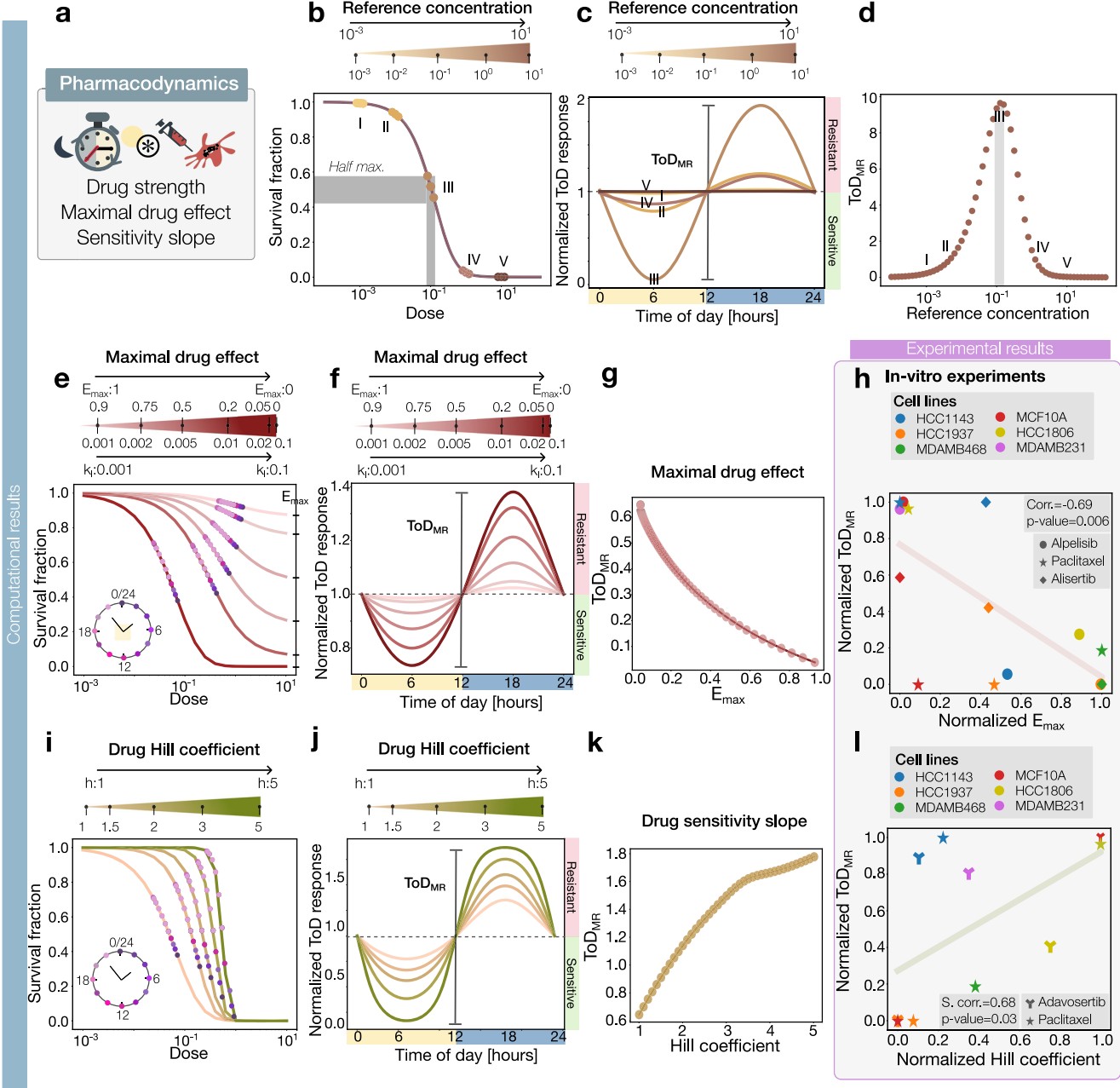

**Fig. 3 | Drug properties modulate the time-of-day drug response. a** Sketch of the drug properties studied in this figure. **b** Simulated survival curve for different dose concentrations shown in the color gradient. **c** Simulated Time-of-Day response curve for the different dose concentrations corresponding to (**b**). **d** Maximum ranges of the ToD response curve from (**b**, **c**) for different reference dose concentrations. **e** Simulated survival curve for different values of the maximal drug effect $E_{max}$ and killing rate. The dots on the survival curve represent drug administration at different times within a day. **f** Simulated ToD response curve for the different $E_{max}$ and killing rate values with the definition of the maximum range. **g** Maximum ranges of the ToD response curve from (**f**) plotted against the $E_{max}$ values. **h** Maximum range of the experimental ToD response curve for the normalized $E_{max}$ values of different cell lines (legend on top). Different symbols correspond to different data sets of the following drugs: Alpelisib, Paclitaxel, and Alisertib. The data was normalized within

each dataset and the Spearman correlation coefficient with the corresponding *p*-value were computed for the entire data collection. The Pearson correlation coefficient with the corresponding p-value is −0.68 and 0.01. **i** Simulated survival curve for different Hill coefficients. **j** Simulated ToD response curve for the different Hill coefficients with the definition of the maximum range. **k** Maximum ranges of the ToD response curve for the different Hill coefficient values. **l** Maximum ranges of the experimental ToD response curve versus the normalized Hill coefficients for different cell lines (legend on top). Different symbols correspond to various data sets of the following drugs: Adavosertib and Paclitaxel. The data was normalized within each dataset and the Spearman correlation coefficient with the corresponding *p*-value were computed for the entire data collection. The Pearson correlation coefficient with the corresponding *p*-value is 0.57 and 0.08.

fractions and resulted in dose-response curves with increased enhanced cancer cell killing rate (Fig. 3e). This, in turn, produced ToD curves with increasing amplitude. Calculating the ToD for each case revealed an inverse proportionality between the maximum range of the ToD response curve and the maximal drug effect $E_{max}$ (Fig. 3f, g).

Considering these computational results, we re-examined previous time-of-day in vitro experiments with chemotherapeutic drugs across different tumor cell lines. For this analysis, we selected drugs with the highest correlation coefficients (Alpelisib, Paclitaxel, and Alisertib) to demonstrate the inverse relationship between the normalized maximum range of the

ToD response and the experimentally calculated maximal drug effect $E_{max}$ for the corresponding drug (Fig. 3h). This analysis shows a weak but significant correlation of −0.68 (p-value = 0.01), suggesting that higher variability in the ToD drug response can be achieved by decreasing the maximal effect of the drug or increasing the maximal killing effect, even when the reference dose is maintained at half-maximal drug concentrations.

Another aspect impacting the time-of-day drug response is the Hill coefficient, which quantifies the cooperativity of drug binding to its receptor. This coefficient indicates how sensitively a drug activates or inhibits its target, affecting the sensitivity to drug concentration (curve steepness) of the response curve as drug concentrations change. In our model, increasing the Hill coefficient led to a rightward shift and greater sensitivity to drug concentration in the survival curves for a cytotoxic drug (Fig. 3i). We also observed that higher Hill coefficients caused broader ranges in the ToD response curve (Fig. 3j), and as the Hill coefficient increased, the maximum range of the ToD response curve expanded, showing a change of slope at higher values (Fig. 3k). From prior experimental assays across different tumor cell lines, we selected drugs like Adavosertib and Paclitaxel, which demonstrated a strong positive correlation between the maximum range of the ToD response curve and the Hill coefficient. This correlation suggests that a higher sensitivity of the drug to concentration variations leads to increased variability in the response (Fig. 3l).

For cytostatic drugs, increasing the maximal inhibitory effect led to similar observations: a rightward shift and enhanced sensitivity in the survival curves (Supplementary Fig. S2b, c). However, variations in the Time-of-Day (ToD) response curve were less pronounced when accounting for a shorter cell doubling time, reflecting the influence of cell growth rate on drug efficacy (Supplementary Fig. S2d). These findings underscore the

importance of considering factors such as the drug's effective concentration, maximal effect, and the sensitivity slope to accurately assess time-of-day drug sensitivities. This relationship illustrates how drug properties interact with circadian biology, potentially affecting the optimal timing and dosing of treatments.

## Drug stability shapes time-of-day drug responses

Time-of-day assays typically utilize the same drug dose administered to a population of cells at different times throughout the day. Experimentalists can either prepare a fresh drug preparation at each time point or use a master drug preparation for all time points. The first approach can induce artificial results due to preparation-to-preparation variability, while the second approach can lead to decreasing drug efficacy due to storage-associated effects. To investigate the latter, we studied the efficacy of drugs with different half-lives.

To better understand how drug stability affects its effectiveness at different times of the day, we developed a mathematical model that includes an exponential decay function to simulate how drug effects decrease over time. Our model examines a wide range of drug stabilities, from drugs that are relatively unstable with a short half-life of 4 h, to stable drugs that do not degrade at all (infinite half-life), as shown in Fig. 4a.

Using this model, we explored how the stability of a cytostatic drug influences its ability to inhibit cell growth when administered at its maximum effective concentration ($E_{max}$). We refer to the elapsed time between when a drug is prepared and when it is administered as the "preparation age."

Our findings reveal a clear inverse relationship between the survival of the cell population and the preparation age of the drug, particularly evident

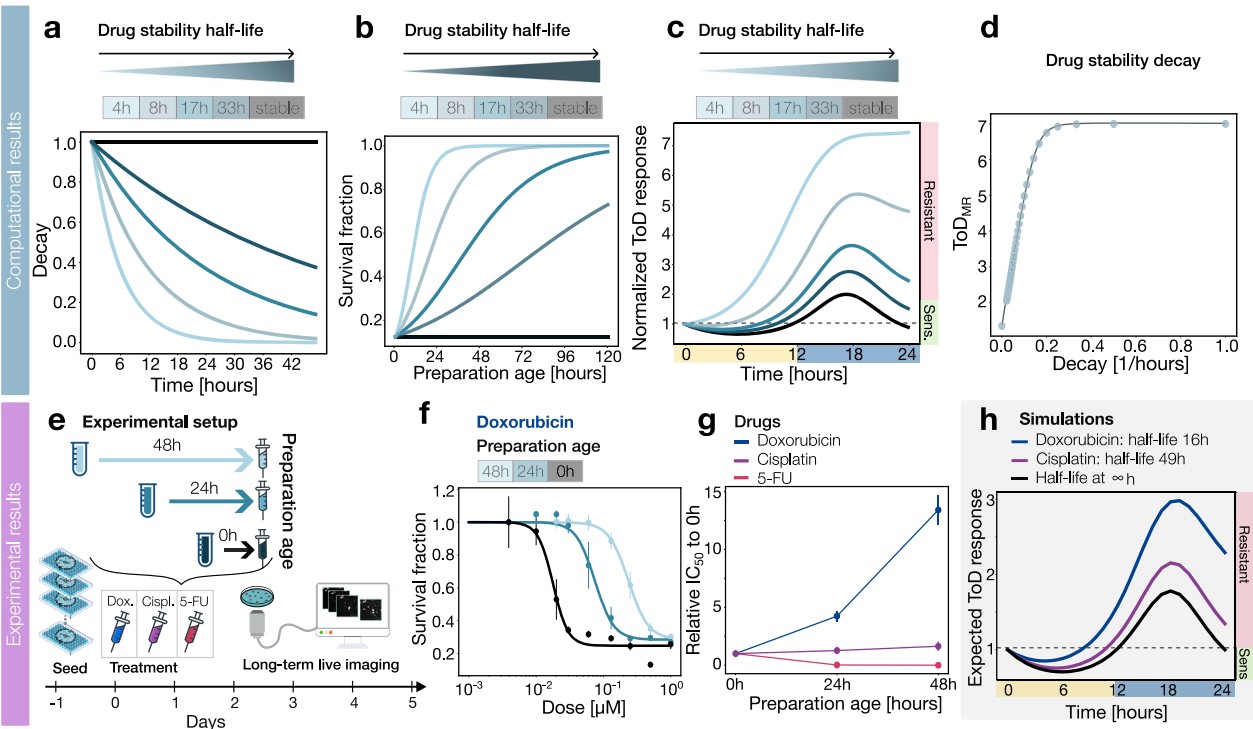

**Fig. 4 | Drug stability strongly affects the time-of-day drug response. a** Drug stability function for different decay values. **b** Simulated survival fraction as a function of the time since drug preparation, preparation age, for the drug stability with different drug stability values (see Methods). **c** Simulated ToD response curve for drugs with different drug stability values shown in the legend. **d** Maximum range of the simulated ToD response curve for different drug stability values of the drug. **e** Experimental setup showing different preparation ages (0, 24, and 48 h) for the following drugs: Doxorubicin, Cisplatin, and 5-FU. The cells were seeded 24 h before treatment with the drugs prepared on different days. Then, the cells were monitored

for 5 days with live imaging. **f** Experimental survival fraction corresponding to Doxorubicin for different preparation ages: 0, 24, and 48 h. The points represent the mean value and the error bars the standard deviation derived from three technical replicates. **g** Experimental relative $IC_{50}$ to 0 h for Doxorubicin, Cisplatin, and 5-FU according to the preparation ages: 0, 24, and 48 h. The points represent the mean value, and the error bars represent standard errors of the residuals from the curve fit of (**f**) (see Methods). **h** Simulated ToD response curve with the experimental decay values obtained from the fitting of (**g**) for Doxorubicin and Cisplatin compared to the simulated ToD response curve for a stable drug.

in drugs with longer half-lives. This means that drugs with longer half-lives, which degrade slower, tend to be more effective over longer periods. However, as these drugs age, their effectiveness gradually decreases, leading to a higher survival fraction among the treated cells, as illustrated in Fig. 4b.

Next, we explored how different stability conditions might affect the time-of-day (ToD) response profiles of drugs. We found that drugs with shorter stability lost the oscillatory characteristic typically seen in ToD drug sensitivity. This led to a broader range of resistance and more consistent drug responses in the second half of the day (Figs. 4c and Supplementary Fig. S3a). As a result, the maximum range of the ToD response curve increased exponentially in cases of shorter stability (Fig. 4d).

To experimentally evaluate these observations, we conducted a drug response analysis on cells treated with three of the most used chemotherapeutic drugs: the antitumor antibiotic Doxorubicin, the alkylating agent Cisplatin, and the antimetabolite 5-Fluorouracil (5-FU). These chemotherapeutic agents induce anti-tumor effects through different mechanisms by interfering with DNA repair, replication, and nucleotide synthesis. We prepared the drugs at different time points before their administration to assess the impact of drug stability (Fig. 4e).

We compared the population survival of three drug preparations (0, 24, and 48 h). Consistent with our simulations, we observed a shift of the survival curves to higher concentrations as the preparation age of Doxorubicin increased, indicating reduced drug efficacy (Fig. 4f). On the other hand, the survival curves for Cisplatin and 5-FU appeared relatively similar across different preparation times, suggesting stable efficacy over the observed period (Supplementary Fig. S3b).

By increasing the preparation age, the $IC_{50}$ value of Doxorubicin exhibited an exponential increase, whereas the $IC_{50}$ values of Cisplatin and 5-FU remained relatively stable (Fig. 4g). To quantify the drug stability decay of these drugs and compute the expected ToD response we fitted the relative $IC_{50}$ as a function of the preparation age (see Methods). Doxorubicin exhibited the highest drug stability decay rate, resulting in an absence of oscillatory behavior in the second half of the day and a predominance of the resistant phenotype, consistent with our simulation results (Fig. 4h). Conversely, we observed a restored oscillatory ToD response curve for Cisplatin with a smaller drug resistance range, which had a lower decay rate (Fig. 4h). Altogether, our results underscore the importance of considering, drug stability and preparation age in the experimental design of time-of-day drug assays.

## Continuous cell growth influences the time-of-day drug response

In the vast majority of ToD assays, all cells are seeded at an initial timepoint and continuously proliferate during the experiment, resulting in distinct cellular conditions at each drug administration time point throughout the day. To study the effect of growing cell population in drug responses we simulated the effect of a population of cells with different doubling times that are treated a 4 h intervals within 24 h, which resulted in significant differences in their cell counts at the time of treatment (Fig. 5a). To quantify the effect of growing cell population on ToD responses, we computed the time-of-day drug response relative to the initial time point of treatment. Our model showed an asymmetry in the second half of the day with higher variability obfuscating the response dynamics suggesting drug instability (Fig. 5b). For cytostatic drugs, these drug responses changed with the cell division rates, having bigger differences for highly proliferative cells. However, when considering the ToD drug response relative to the treatment time and therefore considering the cell density when the drug was administered, we recovered the expected symmetry (Fig. 5c). Notably, higher doubling times generated broader ranges of ToD drug responses.

Considering these computational findings, we conducted experiments using ten populations of cells, each treated at a distinct time point within a day with Doxorubicin, Cisplatin, and 5-FU. We employed a fully automated programmable pipetting system to ensure precise drug administrations at 3-h intervals in a time window of 27 h (Fig. 5d). Here, we observed distinct

proliferation rates and growth dynamics among cells treated at different time points, regardless of the drug type (Supplementary Fig. S4a).

Next, to analyze the impact of cell growth on the ToD drug response, we quantified the relative change in cell number observed 5 days after treatment when considering the initial or the treatment time (Fig. 5e). As expected from the simulations, the ToD sensitivity range across the distinct drugs showed an oscillatory fluctuation for the different treatment time points of the day. When considering the treatment time as the referential time point, we observed that the resistant phenotype range was reduced for Doxorubicin and Cisplatin. In the case of 5-FU, the oscillatory pattern weakened, and the resistant range decreased significantly enhancing the sensitive phenotype (Supplementary Fig. S4b). In summary, these results manifested how the drug sensitivity outcome can be masked by considering the cell density of the incorrect referential time point.

## Assay length influences the time-of-day drug response

To assess how the experimental length affects drug response dynamics, we simulated population growth treated with different doses (Fig. 5f) and measured survival fractions at equidistant time points over 5 days (Fig. 5g). Notably, the survival fraction curves shifted towards lower concentrations and expanded their effective range with longer evaluation periods. The sensitivity slope of the survival curve decreased with the evaluation time, suggesting a more gradual response at earlier evaluation time points. Parametrizing the survival curves, we fitted across the evaluation times to obtain the $IC_{50}$, $E_{max}$, and Hill coefficient (see Methods). For later evaluation times, the relative $IC_{50}$ and $E_{max}$ values exhibited an exponential decrease, ultimately converging toward their asymptotic values (Fig. 5h). The doubling times changed the time required to reach these asymptotic values (Supplementary Fig. S5a–c).

To further determine the assay length effect on the drug sensitivity within a day, we accounted for the circadian modulation of the stress and simulated the response for different time points. Our computational study revealed that time-of-day drug sensitivity exhibited an expanded range of oscillation for later evaluation times (Fig. 5i). This observation implies that extended time lapses are required to fully comprehend the dynamics of drug sensitivity.

Treated cells with Doxorubicin, Cisplatin, and 5-FU were recorded with long-term live imaging to characterize the evaluation time impact on drug response and compare it with our simulation results (Supplementary Fig. S5d). We quantified cell growth by tracking the number of cells over 5 days after treatment (Fig. 5j and Supplementary Fig. S5e). To determine the drug response, we computed the survival fraction within the tested dose range for the three drugs (Fig. 5k and Supplementary Fig. S5f). We computed individual survival fraction curves for four distinct evaluation time points after the treatment: 48, 72, 96, and 120 h. As expected from the simulations, the shape and overall range of the survival fraction curves changed with to the evaluation time points. For short evaluation times, the range of drug response decreased for all tested drugs changing considerably the assessment of the overall drug effect.

To evaluate the impact of assay length, we computed the values of $IC_{50}$, $E_{max}$, and the Hill coefficients for the resulted survival curves, normalizing them to their respective values at 48 h for comparison (Fig. 5l and Supplementary Fig. S5g–l). Notably, the relative $IC_{50}$ values of Doxorubicin and Cisplatin exhibited an exponential decrease as the evaluation time progressed towards later time points (Supplementary Fig. S5g). Since cells treated with 5-FU did not reach 50% growth inhibition at 48 h, the relative $IC_{50}$ value for subsequent evaluation times could not be calculated. The $E_{max}$ values decreased with the evaluation time for all three drugs (Supplementary Fig. S5h). The Hill coefficients showed increasing trends as the evaluation shifted to later periods, as already predicted from the simulations (Supplementary Fig. S5i).

To connect our experimental observations with simulations, we chose Cisplatin due to its highly variable drug response across the evaluation times. We calculated the expected ToD response for the drug parameters derived from the survival fraction curves fed in our mathematical model

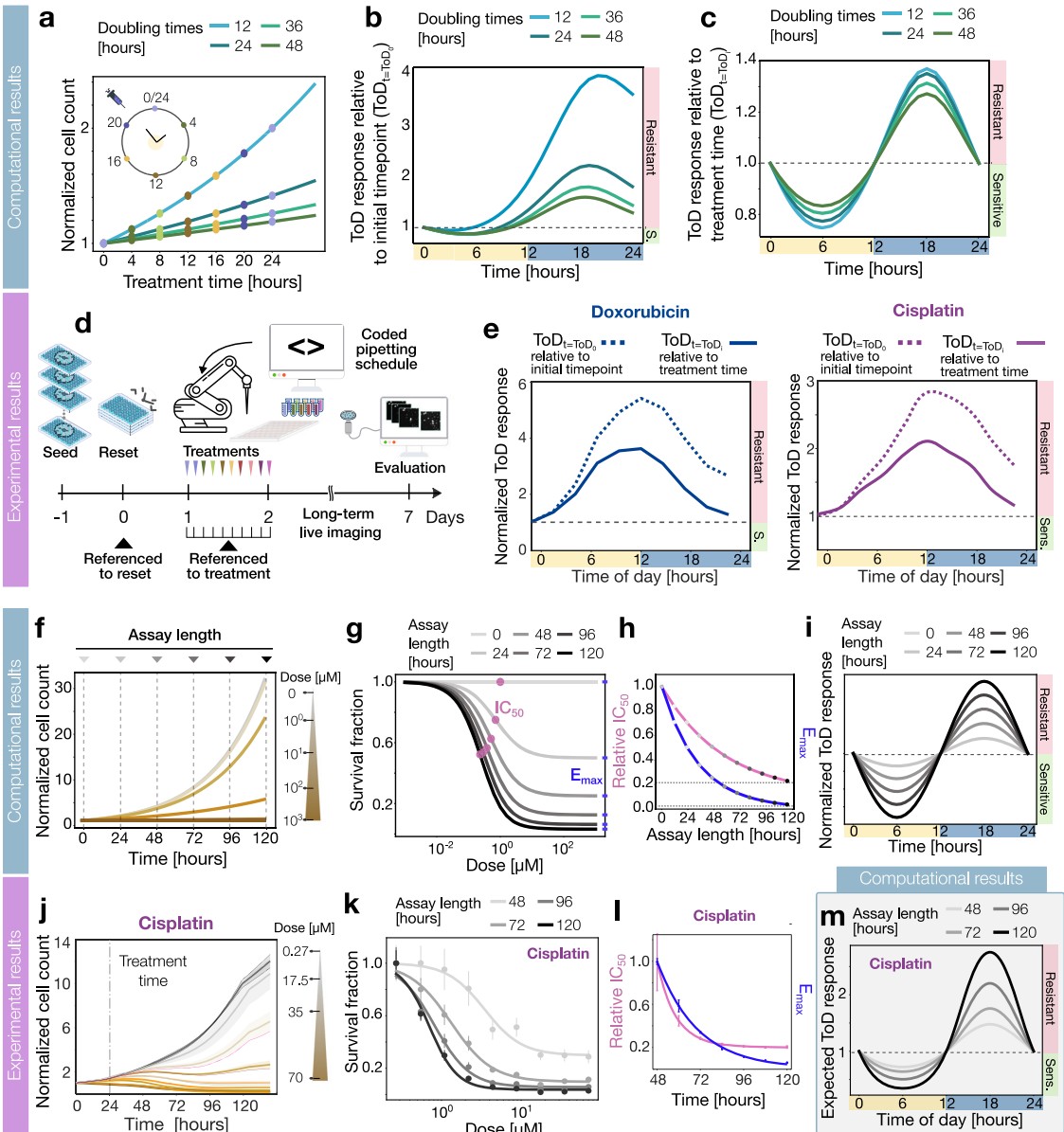

**Fig. 5 | Cell growth dynamics and assay length modify the time-of-day drug response. a** Simulated normalized cell counts to the first time point where each dot shows the treatment administration each 4 h for different doubling times. **b** Simulated ToD response curve with the circadian clock modulation relative to the initial time point for different doubling times. **c** Simulated ToD response curve with a circadian clock modulation relative to the time point of treatment for different doubling times. **d** Experimental setup to study the effect of cell growth dynamics on the ToD drug response. **e** Experimental ToD response curves for Doxorubicin, Cisplatin, and 5-FU relative to the initial (dashed line) or treatment time (continuous) points. **f** Simulated normalized cell counts to the first time point for different doses (gray to brown color gradient) with the assay lengths indicated as vertical dashed lines. **g** Simulated survival curve for different experimental lengths from 0 to 120 h (gray to black gradient color) with the definition of $IC_{50}$ (pink) and $E_{max}$ (blue). **h** Relative

half-response concentration $IC_{50}$ (pink) and maximal drug effect $E_{max}$ (blue) for different assay lengths from 0 to 120 h obtained from the fitting of (**g**). **i** Simulated ToD response curve for different assay lengths from 0 to 120 h. **j** Normalized cell counts to the first time point for different doses (gray to brown color gradient) for Cisplatin. **k** Experimental Cisplatin survival curve for different assay lengths from 48 to 120 h (gray to black gradient color). The points represent the mean value and the error bars are the standard deviation derived from three technical replicates. **l** Experimental relative $IC_{50}$ (pink) and experimental $E_{max}$ (blue) for different assay lengths from 48 to 120 h obtained from the fitting of (**k**). The points represent the mean value, and the error bars represent standard errors of the residuals from the curve fit of (**k**) (see Methods). **m** Simulation of the expected ToD response curve of Cisplatin for different assay lengths from 48 to 120 h with the parameters obtained from the fitting of (**k**).

(Fig. 5m). As predicted in Fig. 5i, longer evaluation times resulted in greater fluctuations in ToD drug sensitivity. Therefore, our findings underscore the role of the experimental length in precise evaluation of the time-dependent pharmacological responses.

## Discussion

Pioneering studies have demonstrated the potential of cancer chronotherapy as a translational field, highlighting the relevance of the circadian

clock in central cancer processes such as tumor initiation, progression[52], invasion[53], and treatment-related side effects[54]. Despite these advancements, the underlying mechanisms and applicability of chronotherapy for different cancer types and drugs are still not fully understood. Several in-vitro assays have investigated how drugs act differently throughout a 24-h day, showing that specific times of day drug administration can significantly affect the response[55]. However, these previous drug screening approaches often neglect the presence of multiple time-dependent confounding factors,

which can severely limit and bias the outcomes. Therefore, it is crucial to investigate how these factors shape the time-dependent dynamics of drug response.

In this study, we investigated the influence of both biological and experimental factors on time-dependent drug response dynamics, examining the roles of circadian clock properties, pharmacodynamics, cell growth dynamics, and evaluation time. These elements were integrated into a unified mathematical model, which was informed and tested using both existing datasets and our novel experimental data. Specifically, existing datasets were utilized to compare the predictions of our model. Additionally, we conducted new experiments (Figs. 4 and 5) to gain insights into how factors such as drug stability and exposure duration directly impact time-of-day (ToD) response curves. This comprehensive approach not only deepens our understanding of drug response dynamics throughout the day but also informs the optimization of time-dependent therapeutic strategies based on our enriched dataset.

Our modeling strategy is designed to generically describe the effects of circadian modulation without distinguishing whether it is driven by exogenous or endogenous oscillators. Instead, we model the effective impact of this modulation as an oscillatory dose-concentration embedded within a dose-response curve function. This approach provides a straightforward and clear framework that enables us to capture the key features of time-of-day effects. However, it is inherently limited in offering mechanistic insights into how circadian clock modulation operates, as more complex models, such as those presented by Hesse et al.[56] might provide. The primary advantage of our strategy lies in its ability to elucidate the distinct effects of oscillation properties and their interaction with drug characteristics on time-of-day response curves, without necessitating detailed assumptions or speculations about the underlying mechanisms.

Using this approach, our simulations showed that circadian rhythm properties affect time-of-day drug sensitivity, resulting in varying ranges of resistance or sensitivity. Specifically, increasing the amplitude of the circadian clock enhanced the maximum range of the time-of-day drug response, in contrast to changes in the period or decay rate of the clock. Examining common chemotherapeutical drugs in tumor human cell lines, we found that Torin2, Cisplatin, Doxorubicin, and Alpelisib followed trends consistent with our mathematical model.

We found that increasing dose concentration gradually expands the oscillatory range of time-of-day drug sensitivity, reaching a peak at the concentration of the half-maximal drug effect, beyond which the range diminishes. This indicates that, for a reference drug concentration, higher doses do not inherently correspond to an increased amplitude of the ToD sensitivity curve. Instead, the model identifies an optimal concentration that maximizes the ToD curve amplitude. Additionally, our results highlight that drugs with sufficient potency induce significant variations in ToD responses, whereas drugs with lower efficacy—defined by reduced killing rates—exhibit negligible ToD differences. For highly sensitive drugs, characterized by a high Hill coefficient, the variation in time-of-day drug response increases, resulting in more pronounced differences based on the timing of drug administration. We found that human cell lines treated with Alpelisib, Paclitaxel, Alisertib, and Adavosertib exhibited trends consistent with our mathematical model, unveiling the relevance of considering pharmacodynamical factors in the time-dependent drug response experiments.

In our study, the efficiency of lentiviral transfection is a crucial factor that could potentially influence the variability of luciferase reporter expression. However, we address this concern by utilizing non-clonal cell lines and conducting our analyses at the population level. This strategy effectively minimizes the impact of any single, potentially inefficient transfection event. By averaging the results across a population of cells, we ensure that our data reflects a more consistent and robust measure of luciferase activity, thus providing a reliable basis for assessing circadian rhythms. This methodological choice strengthens the integrity of our

findings, mitigating the influence of individual variations in transfection efficiency.

Our mathematical model showed that quickly decaying drugs exhibit no characteristic oscillations in drug sensitivity for treatments over a day, resulting in a drug response saturation within the second half of the day. Evaluating the drug response dynamics of treated cells indicated that the preparation age influenced the survival curves of unstable drugs, such as Doxorubicin, decreasing the degree of growth inhibition as the preparation age increased. However, the preparation time did not lead to significant changes in stable drugs such as Cisplatin and 5-FU. While we showed this for three mainstream chemotherapeutic drugs, similar trends are expected for other drugs. Thus, in experimental protocols with drug administration at different time points from a single stock, considering the drug half-life is crucial.

Moreover, we observed that the continuous population growth of cells influences the outcomes of time-dependent drug sensitivity analysis: the ToD response increases with the division rate of cells. To address this effect, we incorporated a cell count normalization strategy: referencing those at the time of treatment. While the overall fluctuations in drug sensitivity across different times of the day remained consistent between the classical analysis and the adjusted approaches, the inclusion of continuous population growth dynamics revealed more symmetric patterns in the time-of-day drug response. Therefore, if the drug response is analyzed without considering the correct referential time, the outcome can be obscured, potentially leading to incorrect conclusions. These findings underscore the importance of accounting for continuous cell growth when analyzing time-dependent drug sensitivity, particularly when considering the reference time point.

Our results showed long assay lengths increased the range of drug responses. Furthermore, the $IC_{50}$ and $E_{max}$ values exhibited an asymptotical behavior as the experimental length increased. The results from the experiments with Doxorubicin, Cisplatin, and 5-FU suggest that drug response evaluations conducted at later time points, beyond the time point of $IC_{50}$ saturation, provide more reliable and consistent data, while evaluations at earlier time points may yield higher variability. Therefore, selecting an optimal evaluation time after $IC_{50}$ saturation is crucial for an accurate and reliable assessment of drug response dynamics.

It is important to note that in this study we use damped cosine functions to represent circadian modulation but that not all circadian oscillations are best described by cosine waves. Periodic rhythms may exhibit asymmetric rise and fall phases or other complex patterns that are not well-captured by cosine functions. State-of-the-art modeling approaches, such as Gaussian Processes[57,58], offer greater flexibility in representing diverse oscillatory behaviors and could provide additional insights into the dynamics of circadian modulation. The findings presented here may be influenced by the choice of oscillation model, and employing more flexible representations might alter specific results. Future studies could explore the impact of alternative model classes on time-of-day effects.

In this study, we modeled the time-of-day effects on drug response based on the conventional 24-h circadian observation cycle. It is crucial to note, however, that actual circadian period can vary significantly across different biological systems and even within the same organism under varying conditions. Our results show that a standard 24-h framework can impact the shape and interpretation of ToD curves, especially when the intrinsic circadian period deviates from this norm. On the other hand, adapting the evaluation ToD timeframe to match the intrinsic circadian period of the system under study can lead to more accurate ToD curves. Consequently, we recommend that experiments aimed at assessing ToD effects consider adjusting their observation periods to align with the intrinsic circadian rhythms of their samples, rather than defaulting to a fixed 24-h cycle. This approach ensures that the resulting data more faithfully represents the physiological realities of the model, potentially leading to more effective and tailored therapeutic interventions.

Here we explored the efficacy of drugs on tumor cells based on a presumed stable circadian cycle. Our simplified model has provided a foundational analysis of circadian impacts on drug responses, illuminating

key time-dependent drug dynamics. However, the dynamic nature of tumor circadian rhythms, which may shift in response to treatment and other factors, can complicate the application of these findings to clinical settings. Tumors may not adhere to a fixed circadian period, leading to potential misalignments between predicted and optimal treatment timings as treatment time progresses. Such misalignments can result in decreased drug effectiveness and increased side effects. Moreover, our model does not address the complexities of how drug dynamics might change with ongoing drug exposure and varying cellular states over time. Repetitive treatment doses could induce phase shifts in the tumor's circadian rhythm, necessitating adjustments to treatment schedules. This highlights the limitations of relying on static ToD curves, which do not accommodate the fluctuating circadian dynamics observed in practice. To address these challenges, future research should focus on developing more comprehensive models that incorporate these dynamic factors. Such models would dynamically adjust to the changing circadian rhythms of tumors throughout a treatment regimen, providing a more realistic representation of drug effects throughout the circadian cycle. This approach would better align treatment schedules with the physiological state of each patient, potentially improving therapeutic outcomes.

In conclusion, our findings underscore the importance of integrating circadian rhythms and drug-specific properties when designing treatment schedules to optimize therapeutic outcomes. While evidence continues to grow that circadian medicine can deliver potential health benefits compared to standard care[5,36], its adoption in clinical practice remains limited. This is largely due to the challenge of reconciling treatment timing with clinical priorities, such as safety and efficacy, and the incomplete understanding of the underlying mechanisms.

Our study addresses these challenges by combining experimental insights with mathematical modeling to examine the biological and experimental factors influencing time-dependent drug sensitivity in cancer cells. Although our results are not yet directly translatable to clinical protocols, we emphasize the importance of incorporating these factors into experimental designs. This approach is essential for advancing personalized treatment timing, with the potential to improve therapeutic efficacy and minimize toxicity in chemotherapies and immunotherapies alike.

## Methods
### Materials and experiments
**Cell lines and cell culture.** The source and the culture details of the cell lines HCC1143, HCC1806, HCC1937, HCC38, MDAMB468, CAL51, MDAMB231, MDAMB436, SUM149PT, MCF10A, MCF7, BT459, and U2OS can be found in ref. 37.

U2OS cells expressing nuclear markers (EF1α-mKate2-NLS-Puromycin) were cultured in RPMI media (Gibco) supplemented with 10% fetal bovine serum (FBS, Gibco) and 1% penicillin-streptomycin (Pen-Strep, Gibco). For live imaging microscopy, we used FluoroBrite (Gibco) imaging media supplemented with 10% FBS, 1% Pen-Strep, and 300 mg/l l-glutamine (Gibco). To reduce the evaporation of media during the acquisition, we filled the outer wells of the imaging plate with phosphate buffer saline (PBS, Gibco). To reset the circadian clock in cell populations, they were exposed to a 30-min pulse of 1 μM Dexamethasone (Sigma-Aldrich) 24 h after seeding. Subsequently, cells were washed with PBS and cultured in FluoroBrite for imaging. These cells were seeded to either 96-well plates (Corning) or flat-bottom 96-well plates (Ibidi). The seeding density in drug response experiments was 750 cells/well and 1000 cells/ well for time-of-day treatment experiments.

Cells were split regularly to maintain their ability to divide and proliferate. They were cultured in a controlled 37 °C and 5% $CO_2$ environment and routinely tested for mycoplasma for quality assurance. All cell lines were monitored for morphology, growth characteristics and health, mostly by long-term live-cell imaging, confirming that cell line-specific features did not vary throughout the study. The cells were authenticated using SNP-typing technology, Multiplexion GmbH.

**Drug preparation.** Drug preparation of the cell lines HCC1143, HCC1806, HCC1937, HCC38, MDAMB468, CAL51, MDAMB231, MDAMB436, SUM149PT, MCF10A, MCF7, BT459, and U2OS can be found in ref. 37.

For the drug response experiments with U2OS cells expressing nuclear markers, serial log2-dilutions of 5-fluorouracil (5-FU Sigma-Aldrich) and Doxorubicin (Hölzel) were freshly prepared in DMSO (Sigma-Aldrich), and for Cisplatin (Sigma-Aldrich) in 0.9% NaCl ultrapure water. The dilutions were further diluted with RPMI media to achieve the desired concentrations. Considering the effect of DMSO on cell growth, each dilution was prepared so that the final concentration (v/v%) of DMSO did not exceed 0.5%. Tested drug concentrations ranged from 100 to 0.39 μM for 5-FU, 70 to 0.27 μM for Cisplatin, and 1 to 0.004 μM for Doxorubicin. Untreated controls were prepared with solvent and media.

For the drug stability analysis with U2OS cells expressing nuclear markers, the drug dilutions mentioned above were prepared at 3 different time points: 2 days, 1 day, and right before the treatment. For each dilution, 200 μL of drug mix was transferred to a 96 well plate (Corning) and stored at room temperature until the treatment. To prevent evaporation of the media and the solvent from the drug dilutions, the 96-well plate was covered with a sheet of aluminum foil during storage.

For the time-of-day treatment experiment with U2OS cells expressing nuclear markers, 1.8 μM of 5-FU, 1.8 μM of Cisplatin, and 0.0023 μM Doxorubicin were prepared. For each drug, untreated controls composed of solvent and media were prepared. The drugs were transferred to a 96 well plate and stored on the tray where the robotic pipette arm reaches during the time-of-day experiment.

**Microscopy.** The setup of the circadian bioluminescence recordings and continuous live-cell monitoring of the cell lines HCC1143, HCC1806, HCC1937, HCC38, MDAMB468, CAL51, MDAMB231, MDAMB436, SUM149PT, MCF10A, MCF7, BT459, and U2OS can be found in ref. 37.

Long-term live imaging of U2OS cells expressing nuclear markers for the drug response experiments was accomplished by an incubator-embedded microscope (Incucyte, Sartorius). Images were acquired in the widefield and red fluorescent channel (excitation: 567–607 nm, emission: 622–704 nm) every hour at 4X magnification for 7 days. Frame-by-frame cell counting was accomplished by the Incucyte image analysis software and the results were further processed in MATLAB. Imaging of cells started after the cells attached to the plate bottom, approximately 2 h after seeding. The cells were taken out from the Incucyte for drug administration and placed back to the instrument for the following acquisitions.

For the time-of-day treatment experiments of U2OS cells expressing nuclear markers, we used a high-resolution widefield and confocal microscope (ImageXpress-micro, Molecular Device). Different magnifications were applied for acquisition depending on the well plate. 10X magnification was applied for the acquisition in 96 well plates, acquiring 4 sites per well. Images were acquired every hour in widefield, and red fluorescent channels until we tracked cell growths for 5 days after the last treatment. Counting cells from acquired images was accomplished by running a frame-by-frame counting of nuclei via the ImageXpress-micro specific software (MetaXpress).

**Time-of-day treatment.** The time-of-day treatment setups of the cell lines HCC1143, HCC1806, HCC1937, HCC38, MDAMB468, CAL51, MDAMB231, MDAMB436, SUM149PT, MCF10A, MCF7, BT459, and U2OS can be found in ref. 37.

Drugs prepared for the time-of-day treatment of U2OS cells expressing nuclear markers were stored in a 96-well plate (Compound plate) and placed on a pre-defined site within the instrument which is reachable by the robotic pipette arm. The pipetting operations were programmed in the instrument-specific software, MetaXpress. We programmed the pipetting sequence in a sequence that the first treatment takes place 24 h after resetting the circadian clock of the cells. From the first treatment on, the subsequent cultures were treated sequentially in 3-h intervals for an overall treatment

time-window of 27 h. For each pipetting event, a solvent-only treated control was assayed alongside.

**Statistics and reproducibility.** Bioluminescence experiments were conducted twice to confirm their reproducibility. We have not derived statistics from the bioluminescence experiments. Live-imaging experiments were carried out a single time because of the comprehensive data generated, according to the standards in the field. The evaluation of each condition was based on a detailed analysis of the captured individual images. Each experiment had three technical replicates. We have derived standard deviation errors from these replicates.

## Computational methods

**Cell growth.** To model cell population growth under the influence of a drug, we adapted the model from[59] and incorporated a circadian modulation. Exponential growth is inhibited by drugs that might have a cytostatic or killing effect as follows:

$$\dot{x} = growth - cytostatic_{effect} - killing_{effect}$$

$$\dot{x} = x * k - x * k * cytostatic_{effect} - x * killing_{effect}$$

$$\dot{x} = x * k \left(1 - cytostatic_{effect}\right) - x * killing_{effect}$$

$$\dot{x} = x * k * \left(1 - \frac{S_m c^h}{SC_{50}^h + c^h}\right) - x * \frac{k_l c^h}{LC_{50}^h + c^h}$$

where $x$ is the cell count, $c$ the drug concentration, $S_m$ the maximal inhibitory effect, $SC_{50}$ the concentration at the half-maximal drug inhibitory effect, $k_l$ the maximal killing effect, $LC_{50}$ the concentration at the half-maximal drug killing effect, and $h$ the Hill coefficient. The untreated division rate per day is computed as $k = \frac{\log 2}{T_{division}}$. Integrating in time, we obtain the cell counts:

$$x(t) = x_o exp \left[t * k * \left(1 - \frac{S_m c^h}{SC_{50}^h + c^h}\right) - t \frac{k_l c^h}{LC_{50}^h + c^h}\right]$$

**Circadian modulation.** To describe the circadian impact on the drug, we considered a sinusoidal modulation of the $IC_{50}$ (the concentration at the half-maximal response):

$$c(ToD) = IC_{50} * circadian_{modulation}$$

$$circadian_{modulation} = m * A + m * A * \sin \left(\frac{2\pi}{T} ToD + \varphi\right),$$

where m is the strength, A the amplitude, T the period, ToD the time where we consider the drug effect and $\varphi$ the phase of the circadian modulation. We sum $m*A$ to the modulation to avoid negative values in the concentration. In all simulations this first term is considered as 1, except when stating otherwise.

The circadian modulation can decay in time exponentially as a damped oscillator:

$$c(ToD) = IC_{50} * circadian_{modulation},$$

$$circadian_{modulation} = sinusoidal_f * exponential_{decay}$$

$$circadian_{modulation} = 1 + m * A * \sin \left(\frac{2\pi}{T} ToD + \varphi\right) * exp(-d * ToD)$$

where d indicates the decay value.

**Concentration.** To describe the effect of the concentration variation, we gradually changed the concentration as follows

$$c_{ToD} = c_i * 10^{m*\left[1+A*\sin\left(\frac{2\pi}{T} ToD + \varphi\right)\right]},$$

where $c_i$ is the considered concentration. To standardize the circadian modulation across various concentrations on a logarithmic scale (log10), we implemented it as an exponential function with base 10. This approach ensures consistently equal modulation around the target concentration, irrespective of its value within the logarithmic scale.

**Drug stability.** To model the drug response considering the preparation age, we described the drug stability as an exponential decaying function

$$c(ToD) = IC_{50}\left[1 + m * A * \sin\left(\frac{2\pi}{T} ToD + \varphi\right)\right] * exp(-d * ToD),$$

where d indicates the decay value. We considered different decay values for Fig. 4: drugs that are stable for $time_{stable}$ = 6 h, 12 h, 24 h, 48 h and ∞h (to consider the ideal case of drugs that do not decay in time). These values transformed in rates as $d = 1/time_{stable}$ = 0.17, 0.08, 0.04, 0.02 and 0 1/h. For the labels in Fig. 4, we used the half-life values which are obtain as $time_{1/2} = \frac{\ln(2)}{d} = \ln(2) * time_{stable} \cong$ 4 h, 8 h, 17 h, 33 h and ∞h.

**Time-of-day response curve.** The population drug response within a day was determined by the ratio of cell counts at the end and the cell counts at the beginning/treatment of the assay/simulation:

$$ToD = \frac{x\left(c, t_f\right)}{x(c, t_o)},$$

$t_f$ is the end time point of the recording/simulation, and $t_o$ is the treatment or initial time. We normalized the ToD response to the initial value to allow for comparable scaling between different conditions.

**Maximum range of time-of-day response curve.** The maximum range was computed as the absolute difference between the maximum and minimum of the ToD response curve: $MR = |\max(ToD) - \min(ToD)|$.

**Survival curve.** The survival population fraction was computed as the ratio of cell counts between a particular concentration and the untreated case at the end of an assay: $SF = \frac{x(c, t_f)}{x(0, t_f)}$.

**Correlation coefficient and _p_-value.** The correlation coefficients and p-values were computed with the formula *pearsonr* and *spearmanr* from the package *Scipy* of Python[60]. For the main figures, we chose the drugs with a correlation coefficient higher than 0.45 and a p-value smaller than 0.05.

**Obtaining circadian parameters from experimental recordings.** We utilized bioluminescence recordings from several cancer cell models extracted from a recent work from our lab, where luciferase reporters under the control of the Per2 and Bmal1 circadian promoters captured the rhythmic gene expression indicative of the cells' internal circadian dynamics[37]. Here, we then analyzed this bioluminescence data by fitting it to an exponentially decaying sinusoidal function, allowing us to extract key circadian parameters such as amplitude, period, and amplitude decay rate:

$$f(t) = A_0 e^{-\gamma t} \sin\left(\frac{2\pi}{T} t + \phi\right)$$

where $A_0$ is the initial amplitude, $T$ is the period, $\phi$ the phase, $t$ the time, and $\gamma$ the decay rate. We considered valid only the fittings with an $R^2$ value of 0.6 or higher. These parameters are presented and discussed in Fig. 2l–n.

**Table 1 | Drug simulation parameters used in the presented figures**

| Figures | $S_m$ | $SC_{50}$ | $h$ | $k_I$ | $LC_{50}$ |
|---|---|---|---|---|---|
| 2a–c, 4a–d, and 5a–c, f–i | 1 | 1 | 1 | 0 | 1 |
| 2e–g, i–k | 0 | 1 | 1 | 0.01 | 1 |
| 3a–c | 0 | 1 | 1 | 0.05 | 1 |
| 3d–f | 0 | 1 | 1 | [0.001, 0.1] | 1 |
| 3h–j | 0 | 1 | [1,5] | 0.1 | 1 |
| S2a, b | [1,10] | 1 | 1 | 0.1 | 1 |
| S2c | 1 | 1 | [1,5] | 0.1 | 1 |

**Determination of drug response parameters**. In Fig. 4g, to determine drug response parameters, we analyzed the IC50 stability decay as a function of the preparation age, fitting the data to an exponential function: $y = a * \exp(d * x) + c$. These decay values were used for the subsequent simulation in Fig. 4h.

The survival curve was fitted to the following formula from[59]:

$$SF = E_{\max} + \frac{1 - E_{\max}}{\left(1 + \frac{c}{IC_{50}}\right)^h},$$

where $E_{\max}$ is the maximal drug effect at an infinite concentration, $h$ the Hill coefficient of the survival curve which reflects its sensitivity to drug concentration, also referred as steepness, and $IC_{50}$ the effective concentration at the half-maximal response. For the simulation from Fig. 5m, we extracted the parameters from the fitting to the above equation. Assuming the final $IC_{50}$ and $E_{\max}$, we simulated the expected time-of-day curve with the experimental parameters.

**Simulation parameters**. The circadian parameters for Figs. 2a–c, e–g, i–k, 3a–c, d–f, h–j, 4a–d, and 5a–c, f–i are: $A = 1$, $T = 24$, $\varphi = 0$, and $m = 0.5$. For all simulations, we used the circadian signal decay $d = 0$, except for Fig. 2g, k.

The drug parameters used are shown in the following table: Table 1

**Reporting summary**

Further information on research design is available in the Nature Portfolio Reporting Summary linked to this article.

## Data availability

All the data generated and used for this investigation is available here[61]: https://doi.org/10.6084/m9.figshare.28423709.

## Code availability

The implemented code used for this study can be found here: https://github.com/Granada-Lab/drivers_ToD_drug_sensitivity.

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

## Acknowledgements
We sincerely appreciate the valuable feedback provided by the members of the Institute for Theoretical Biology, AG Keilholz and AG Kramer from Charité throughout this study. We are grateful for the financial support from the German Federal Ministry for Education and Research (BMBF) through the e:Med Juniorverbund DeepLTNBC TP 3 - 01ZX1917C, which made this research possible.

## Author contributions
Conceptualization and investigation: N.G., H.I., C.E. and A.E.G. Methodology: N.G., H.I., C.E. and A.E.G. Experiments: H.I. and C.E. Visualization: N.G., A.E.G. Funding acquisition: A.E.G. Project administration: A.E.G. Supervision: A.E.G. Writing – original draft: N.G. and A.E.G. Intellectual support: U.K. and H.H.

## Funding

## Competing interests
The authors declare no competing interests.
