## [Transparent Peer Review file · Communications Biology]

A combined mathematical and experimental approach reveals the Drivers of Time-of-Day Drug Sensitivity in Human Cells

Corresponding Author: Dr Adrian Granada

Version 0:

Reviewer comments:

Reviewer #1

(Remarks to the Author)

Please see the attached reviews.

Reviewer #2

(Remarks to the Author)

Gutu and colleagues aim to better understand circadian modulation of drug sensitivity. The authors combine modeling with a re-analysis of a data set they recently published in Nature Communications.

However, when I tried to focus on what was explicitly done/learned in this paper (as opposed to the already published data), it seemed as if the primary methods were to assume a sinusoidal form for various parameters (one at a time) and plot the results of those sinusoidal assumptions. Similarly, some of the other conclusions seemed well known. The language of the paper makes it seem that they did more extensive “simulations” than this and given the author’s well-established expertise I worry that I may have misunderstood. I have tried to highlight points where clarification may help.

General:

Despite the aim to model temporal complexity, the models appear to assume fixed and unchanging drug effects based on the age of the drug and the circadian state of the cells at the moment of drug administration. (Indeed, the analytic solution for $x(t)$ in line 495 is only valid if these parameters are constant over the course of the simulation) For example: the simulated growth curves in 2b (and elsewhere) seem to reflect an assumptions of a constant exponential growth (with different but constant rates depending on the time of initial administration). But the experimental system is closed with drug remaining in the dish over time. The authors exposition would seem to suggest a more complicated model with different effective growth/local killing rates at different circadian times as the simulation advances in time and some fraction of the drug remains in the dish? But I do not think this is what was done?

Most of the experimental correlations shown in the main figures (correlations between circadian parameters and various drug effects (e.g 2m, 2n) seem to be primarily driven by 1-2 points with very high leverage. Moreover, each point is a completely different cell type with many differences in biology beyond the amplitude of a single circadian luciferase reporter. Thus, it is hard to be confident that the differences seen in those 1-2 more extreme cell types are driven by the circadian differences between the cells.

Specific Points:

- (1) The equation in solution for $x(t)$ in line 495 is only valid if the parameters in line 489 are constant. But this is not the case (as I understand it) for a system with circadian variation in activity over the course of the trial. I thought this is what you wanted to model? Please clarify – and highlight in the text if the growth rates/kinetic/drug efficacy terms are held fixed for the duration of each simulation
- (2) In describing the equation on line 500 you write that you are trying to model a sinusoid modulation of the ic_{50} . But the equation says $c(t)$ is equal to the terms on the right hand side? What exactly is $c(t)$ in - for example - the equations in 505 and

(3) In Panel 2F and the related description/discussion you conclude that prolonging the period doesn't affect the ToD response until the period is a little beyond 30 hours. It looks like you simply assumed that the influence at time 0 is fixed – and then only compute the influence of TOD for a 24 hour period. Thus if the peak efficacy occurred after the 24 hour period you evaluated, it would not show up in the way that you assess? Is this correct? If however, you did search over the course of the cells full period - there would be no effect?

It is hard to characterize a tumor state at a given time (e.g tumor state every midnight) if the tumor's entrained period is not 24 hours. Each passing day the tumor's clock would shift relative to our wristwatches. If we do assume that tumor clocks might not have an entrained 24 hour period – then the benefits/drawbacks for timing personalized therapy (as the authors emphasize in their discussion) would likely be measured relative to the longer time period? (and given the authors model would not effect TOD) Is this all correct?. In your in-vitro case if I waited for the second day to do the analysis - I would assume I would get different answers as the relative period has shifted.

(4) A few plots compare estimated circadian features based on dynamic luminescence with drug effect (2m, 2n, 2o,) . Separate estimates based on Per2-luc and BMAL-luc transfections are given in the tables. Which estimates were used in the plots? Why was that one chosen? Was it an average? A quick look at the data suggests that the PER and BMAL amplitudes are poorly correlated. It is not clear why the authors assume that the amplitude of a single circadian reporter reflects the amplitude of the clocks influence on drug effect.

Concretely: Given the apparent dependence of the various correlations of on a few outliers, I would ask that rank based (Spearman) correlations and significance measures be reported instead of Pearson correlations for all these plots (and the plots 3h and i). Relatedly the amplitude of the luciferase signal is (I would assume) strongly dependent on the efficiency of the viral transfection. Was there some method to account for this confounder in the analysis? Please specify? The authors should add to the discussion as to why the amplitude of the given circadian reporter is expected to correlate with the circadian amplitude of the particular drug's effect.

(5) The major outcome of the analysis of drug properties (Figure 3) is (I think) simply that if a drug is sufficiently efficacious – there will be less of a TOD difference if it is administered at either its peak or trough circadian efficacy. In the extreme example – if the drug kills all the cells regardless of administration time – there would be no TOD dependence. Is this the point? Please clarify. If this simple (well-known) fact is what is being highlighted – it is likely obscured by plotting only the TOD variation and not the killing rates at the two most extreme times. Consider adding those plots.

(6) I am hopeful that there are some unique/novel points in Figure 5. This is also where there is new experimental data. However, it appeared like the authors were showing that if you wait a while before adding drug, the cells will have had time to replicate, and if not accounted for, this will influence the computed efficacy. This would seem to be a trivial result. I worry I missed something. Could you please explain the take-a-way beyond this? Perhaps it can be less concise if needed.

(7) Some of the discussion was overstated

For example:

"We observed that alterations in circadian rhythm properties affect time-of-day drug sensitivity, resulting in varying ranges of resistance or sensitivity. Specifically, increasing the amplitude of the circadian clock enhanced the maximum range of the time-of-day drug response, in contrast to changes in the period or decay rate of the clock."

As I understand it, the authors specifically assumed that the core clock amplitude influenced the circadian amplitude of drug responsiveness in their mathematical models. In the data (Torin 2) the correlation the measured clock amplitude and ToD effects was driven 1-2 apparent outliers. I am not sure this was really "observed" so much as assumed in the modeling.

"Recent research indicates that circadian medicine offers significant health benefits over standard care "

Recent work has highlighted the potential. To my knowledge, very few (2-3?) clinical studies in specific disease have shown small gender specific benefits. I would be careful not to overstate this.

"we are the first to highlight the importance of incorporating these factors into experimental protocols"

Several authors have highlighted the importance of drug half-life/stability, pharmacokinetics, and pharmacodynamics. Several in-vitro studies have considered different doses. Many in-vitro and in-vivo therapeutic trials (beyond those in circadian medicine) have worked to characterize the killing effect of a chemotherapeutic based on the number of cells (or tumor mass) when the treatment is administered.

The limits of the study should also be discussed

Reviewer #3

(Remarks to the Author)

In this manuscript, the authors investigate potential drivers for Time-of-Day drug sensitivity in human cells, i.e., drug efficacy and/or toxicity.

Such an investigation falls under the umbrella of chronotherapy, a framework postulating that drug efficacy/toxicity varies according to its administration time, and importantly, to the body's internal circadian rhythms. Indeed, circadian modulation also happens at various levels like cell growth or DNA repair, thus influencing drug effectiveness. Moreover, the different mechanisms involved in how the drugs themselves are processed may be under circadian regulation, impacting drug stability and drug properties. This inherent complexity warrants a careful study to precisely decipher the overall contribution of the circadian clock to drug efficacy. This was achieved by the authors by leveraging a combination of mathematical models and experiments.

As for a general impression, the paper is well-written and smoothly organized. I would say that the structure of the paper itself, a sequence of reported findings regarding tedious computational and biological experiments, felt difficult to read at times. But to add some nuance, I agree with the view of the authors on this careful study being necessary. Directly quoting the paper, "previous drug screening approaches often neglect the presence of multiple time-dependent confounding factors, which can severely limit and bias the outcomes. Therefore, it is crucial to investigate how these factors shape the time-dependent dynamics of drug response." and "A comprehensive, transdisciplinary approach that integrates experimental data with mathematical models is necessary to advance this field".

I only have one comment:

Regarding the way the circadian clock is integrated into the models, exogenous rhythms were employed, using damped cosine functions. A decay parameter, phase, amplitude, and mean level parameters govern these oscillations. One can then investigate the effect that different circadian rhythms may have on Time-of-Day drug sensitivity by varying these parameters. Such an approach is an abstraction of endogenous circadian mechanisms, resulting in small-sized models that are easy to fit, but less actionable: if the cosine amplitude appears to be a sensible parameter, how would one experimentally verify that? E.g., which exact member of the circadian clock should be intervened over? This being said, while there exists work attempting to be "more mechanistic" in the way the circadian clock acts on drug sensitivity [1], these often result in large models, highly nonidentifiable, and difficult to fit, which require large amounts of data from many species. Furthermore, it can also be the case that going into such a level of detail is not even required for drawing impactful conclusions. Given all these reasons, I believe the way the authors formulated the study to be best suited. I would recommend:

- Acknowledging the existence of such works.
- Including an explanation as to why exogenous rhythms were considered rather than endogenous rhythms. Currently, an equation describing circadian modulation is included, but the latter does not come with any justification.
- Mentioning the existence of other approaches for representing exogenous rhythms. (Dampened) Cosine waves have been used to capture circadian modulation for a long time, as they provide a simple approach with somewhat interpretable and actionable parameters (see discussion above). Yet, not all oscillations are best represented using cosine waves, e.g., periodic rhythms can be observed with asymmetric rise and slope parts. State-of-the-art modeling tools for that matter now heavily rely on Gaussian Processes [2,3], and employing a different form for oscillations might change the results. To be clear, I am not asking the authors to carry out again all the analyses using a different model. A sentence mentioning that the findings can be influenced by how flexible the model class is for the oscillations would be welcomed, thus providing some nuance to the findings.

Finally,

It is worth mentioning that all the data used throughout the paper was made available by the authors (as Excel sheets). Moreover, the authors created a GitHub repository containing the code used to generate all figures. The code is clean and readable. I did not run every script, but the few I launched worked fine. I would suggest adding a README.md file, indicating:

- What are the necessary packages to successfully run the code
- In what directory should each data xls file be placed.

Abovementioned references:

[1] Hesse et al. A mathematical model of the circadian clock and drug pharmacology to optimize irinotecan administration timing in colorectal cancer. Computational and Structural Biotechnology Journal (2021)

[2] Sahay et al. An improved rhythmicity analysis method using Gaussian Processes detects cell-density dependent circadian

oscillations in stem cells. Bioinformatics (2023)

[3] Durrande et al. Detecting periodicities with Gaussian processes. PeerJ Computer Science (2016)

Version 1:

Reviewer comments:

Reviewer #1

(Remarks to the Author)

The authors have done a very good job at addressing all of the concerns that were raised. I'm satisfied with their responses and would recommend it be published.

Reviewer #2

(Remarks to the Author)

The authors have sufficiently addressed my concerns.

We sincerely thank the reviewers for taking their time to evaluate our manuscript and for offering valuable feedback. In this response, we have carefully addressed each of the concerns raised, including new analyses in the revised manuscript. For ease of review, we are providing a highlighted version of the manuscript to indicate the changes.

Reviewer #1 (Remarks to the Author):

Summary: This paper attempts to characterize the time-dependent cellular and pharmaceutical factors that influence chemotherapeutic drug efficacy in various human cancer cell lines when administered at various timepoints throughout a 24-hour period using both predictive computational modeling and an *in vitro* experimental approach.

We sincerely appreciate the reviewer's time and effort in providing detailed feedback, as well as their thoughtful organization of comments by section and line numbers. This structured approach has greatly facilitated our revision process, and we are grateful for their thorough review.

Abstract:

Minor Comments:

1. I would challenge the notion that chronotherapy has yet to be fully integrated into clinical practice due to insufficient basic science knowledge concerning why it works. Clinically, the burden of proof needed to implement a therapy is that it is safe and that it actually works, not that the specific mechanism through which it safely works is thoroughly understood. While I appreciate the sentiment that basic biology is valuable in optimizing therapies, it is not a primary challenge that chronotherapy faces insofar as its wide-scale implementation is concerned.

Thank you for your comment. We acknowledge the point you raise regarding the integration of chronotherapy into clinical practice. Indeed, the clinical application of any therapy, including chronotherapy, hinges on demonstrating its safety and efficacy rather than a complete understanding of the underlying mechanisms. Our statement aimed to highlight the potential of enhancing chronotherapy through a deeper mechanistic insight, but we agree that this is not the primary barrier to its clinical implementation. We appreciate this correction and will adjust our abstract accordingly to better reflect the realities of clinical practice (see lines 18-21).

Introduction

Major Comments:

1. Lines 45-46, a citation is needed for each of the circadian modulation mechanisms (immune response, pharmacokinetics, and stress response pathways).

We have now included relevant citations for the circadian modulation mechanisms mentioned, specifically for immune response, pharmacokinetics, and stress response pathways. These references are provided in the revised manuscript to support the statements on lines 45-46.

2. Line 80: The specific characteristics of the circadian clock need to be listed.

We thank the reviewer for their suggestion to clarify the circadian characteristics studied. In response, we re-wrote the whole paragraph including more information about our modeling strategy and analysis. We have explicitly listed the specific circadian clock characteristics, which include amplitude, period, and amplitude decay rate. We hope this clarification addresses the concern. See changes in line 78-94.

Results

Major Comments:

1. Line 98: Define reference concentration. This is defined later in the paper but needs to be clearly stated when introduced.

We thank the reviewer for pointing out the need to define "reference concentration" earlier in the manuscript. In response, we have revised the text on line 98 to include a clear definition of the term when it is first introduced. We hope that this adjustment helps the readers have a proper understanding of the concept from the outset (see lines 100-104).

2. Line 101: The authors state that their approach represents how drug effectiveness varies at different times of day. I'm not sure that this can be definitively stated because 1) the authors do not test any drugs within biological organisms and 2) this is in reference to testing difference concentrations, which is extrapolated into efficacy.

Thank you for your constructive feedback. You are correct in highlighting that our mathematical modeling primarily describes a modulation on drug concentration, not directly on effectiveness. We regret any overstatement in our original text and have revised it to reflect the nature of our approach more accurately.

We have now incorporated a clarification about our modeling strategy. Please see the revised sections in lines 106-108 and 115-123. We appreciate your attention to detail and believe these modifications improve the clarity and accuracy of our manuscript.

3. Any time circadian period is mentioned, it is always way too long. This paper is examining drug efficacy in humans, whose circadian periods can only be 22-26 hours, even in extreme cases.

Thank you for your observations regarding the circadian periods mentioned in our study. Our study employs computer simulations to explore parameters within and beyond normal physiological conditions. The range of values was chosen based on our findings from in vitro tumor cell line models like CAL51 SUM149PT and HCC70 (see Rebuttal Figure 1), which exhibit notably prolonged periods. This approach allows us to investigate the potential impacts of extreme circadian variations on time-of-day (ToD) drug response curves. We have updated the text in the relevant sections to clarify that while periods within the 22–26-hour range are typical; our study deliberately examines also the consequences of deviating from these norms to allow the study of circadian influences under atypical conditions. See changes in lines 142-147.

Rebuttal figure 1. Mean period of the circadian rhythm obtained from the fitting (see Methods lines 674-684) of bioluminescence recordings of different tumor cell lines. Data uploaded to the repository stated in the main text: <https://itbfiles.biologie.hu-berlin.de/index.php/s/Ws3LmoPD4xAK7Zp>.

4. Line 112: What are the ToD values being referred to here?

In the original version of our manuscript, line 112 discussed the asymmetry between the maximum and minimum values of the Time of Day (ToD) response curve. We have now improved our discussion of this asymmetry. We now clarify that although the fluctuations in drug concentration are symmetric, they can lead to asymmetric ToD curves. This asymmetry arises because the responses to these fluctuations are influenced by the exponential nature of cellular growth and the characteristics of dose-response curves. These factors combine to produce disproportionate effects on growth at different times of the day, despite uniform changes in drug concentration. We now clarify this in the main text (lines 127-133).

5. Line 118: The circadian parameters need to be specifically stated.

We have revised our manuscript to provide clearer and more detailed descriptions of the circadian parameters we investigated. Specifically, we now explicitly mention and define the amplitude, period, and amplitude decay rate of the circadian signal. See lines 124-126.

6. Line 124: 1) Please define the threshold referenced here. 2) Define circadian decay signal.

Thank you for your feedback. In retrospect, we recognize that our use of the term "threshold" may have caused potential confusion. We have revised the description to reflect the observed data patterns more accurately without referring to a specific threshold. Moreover, we have now clarified the term, circadian decay signal, which describes how the amplitude of the circadian signal diminishes over time. These changes aim to clarify our results and avoid any ambiguity. The corresponding changes can be found in the updated version of the manuscript, lines 142-144 and 147-149.

7. Line 131: The sentences beginning “Here we extracted from each...” is not clear. Consider rewording this to clearly communicate your point.

Thank you for your feedback on the clarity of our text. We have completely rewritten the paragraph in question to ensure it communicates our points more clearly and effectively. See lines 156-165.

8. Line 129: Please name the specific circadian clock signals recorded.

*We have completely rewritten the paragraph in question including the specific signals recorded, *Bmal1* and *Per2* (see line 158).*

9. Lines 178-194: The stated definition of Hill coefficient is wrong. I would suggest re-examining this entire paragraph.

Thank you for your feedback. We have revised the section in question, ensuring that the Hill coefficient is correctly defined. Additionally, we've modified the language to emphasize the biological implications of our findings rather than the geometric aspects of the dose-response curves and their impact on the Time-of-Day (ToD) response curves. See lines 213-225.

10. I would recommend describing your results in terms of circadian relevance rather than just as a function of graphical transformations. This would make your manuscript more relevant to biologists rather than mathematicians.

Thank you for your suggestion to emphasize the circadian relevance of our results. We have now made several modifications across our manuscript to better highlight how the findings connect to circadian biology, with the hope to make it more pertinent to biologists.

11. Lines 206-207: Wouldn't higher decay rates lead to higher survival? This should be stated more clearly.

We thank the reviewer for their feedback and apologize for any confusion caused by our use of the term "decay rates" in the labels of Fig. 4 and the main text. In the original version of the manuscript, "decay rates" referred to the inverse of the time it takes for the drug to lose its efficacy, effectively describing how quickly this loss occurs. Consequently, the statement in lines 206–207, indicating that higher decay rates lead to diminished survival, was meant to convey that a drug with a shorter stability time (higher decay rate) kills fewer cells, resulting in a higher survival fraction.

To address this potential ambiguity, we have rewritten that whole section and modified the labels in Fig. 4 to "half-life values" for clarity. We updated the text to clarify this relationship in the revised manuscript (new lines 242–246) and expanded the explanation of the connection between decay rates and half-life values in the Methods section (lines 647–651).

12. All simulations with drugs do not include which drug was used in the simulation. Drugs are very different and have different properties, and it needs to be indicated which one is being used for these data.

We apologize for the oversight in not clearly specifying the type of drug used in each simulation. We agree that specifying more direct characteristics of a specific drug could provide more drug-specific insights, as different drugs have properties that can significantly impact our results. However, our approach in this study utilizes a generic modeling framework that broadly categorizes the drugs into cytotoxic or cytostatic types. This allows us to investigate the qualitative and relative effects of such drug categories on time-of-day response curves. Throughout the manuscript and the supplementary figures, we aim to illustrate the application of both approaches, providing ample examples from both families of drugs to demonstrate their distinct impacts.

To clarify the specifics of our simulations, in the figures presented (Figs. 2, 3, and S1), a cytotoxic drug was used, while a cytostatic drug was employed in Figs. 4, 5, S2, S3, S4, and S5. We have now explicitly stated the type of drug used in the relevant sections of the Results corresponding to each figure (see revised lines 114, 186, 196, 226, 247, and 295) to eliminate any ambiguity.

Minor Comments:

1. Lines 110-111: The authors state, “Increasing amplitude of circadian oscillations results in ToD responses with higher amplitudes.” This seems inherently intuitive and does not need to be stated.

We thank the reviewer for their observation and agree that the statement in question appears intuitive. As a result, we have removed it from the revised manuscript (see lines 126–127).

2. Line 127: What are the predictions?

We thank the reviewer for pointing out the misuse of the word "predictions" and apologize for this oversight. In the updated version of the manuscript, we have revised this sentence accordingly (see lines 156–158).

3. Line 156: The phrase “dose strength” is redundant.

We agree with the reviewer that the expression “dose strength” is redundant. Accordingly, we have revised it to "dose" in the updated manuscript (see line 173).

Discussion

Major Comments:

1. Line 331: What does the phrase “stress sensitivity regulation” refer to? Is this what was tested? If so, then how? If not, new concepts relating to the data should not be brought up for the first time in the discussion.

Thank you for highlighting the inconsistency regarding the phrase "stress sensitivity regulation" mentioned in line 395 (former 331). We appreciate your attention to detail. In response, we have removed that statement from the discussion, as it was not directly tested in our study and its inclusion could lead to confusion.

2. Line 369-370: The statement concluding with “...benefits over standard care.” needs a citation.

We have now added relevant references to this and also rephrase that sentence (lines 496-487).

Methods

Major Comments:

1. The authors need to state how they are measuring circadian rhythm parameters as well as define these parameters.

We have addressed this by including a separate subsection in the Methods section of our manuscript, where we explicitly detail the procedures for measuring circadian rhythm parameters. See lines 674-684.

Minor Comments:

1. Line 418: There is some sort of issue with formatting the concentration of dexamethasone. It seems to be 1M dexamethasone, but please clear this up.

We thank you the reviewer for pointing out this formatting issue with 1 μ M Dexamethasone. This has been corrected in the main text (see line 536).

2. Line 503: What is the opposite of 1? Try using a different word like “otherwise” to make this clearer.

We thank the reviewer for noting the misuse of the word "opposite." We have revised it to "otherwise" in the updated manuscript (see lines 626-627).

3. Line 508: “Concentration strength” is redundant.

We agree with the reviewer that the expression “concentration strength” is redundant. Accordingly, we have revised it to "concentration" (see line 634).

4. It might help to show the derivation of the equations used.

We thank the reviewer for their suggestion and agree that including the derivation of the equations would benefit the reader. These derivations have now been added to the revised manuscript (see lines 605-611, 622-623 and 629-631).

Figures

Major Comments:

1. Consider illustrating your computational model alongside experimental results so as to facilitate a greater understanding and appreciation of what it is exactly you are predicting.

During the preparations of this manuscript, we carefully considered various approaches, including integrated visuals (overlying computational data directly onto experimental results) and sequential one-to-one presentation (showing one prediction followed by the corresponding experimental data). However, these approaches often resulted in dense, cluttered visuals that were difficult to interpret due to overlapping lines, colors, and excessive information.

After a few iterations, we finally adopted in many figures a "row-by-row" format, presenting rows of simulation results followed a row of experimental results, clearly distinguishing the two. This method helped us to make a clean and readable presentation, and also allowed us to be explicit about the nature of each result—clearly differentiating computational simulations from experimental data.

Additionally, this approach is coherent with our goal of demonstrating qualitative similarities between the computational model and experimental results rather than aiming for quantitative matching, which is beyond the scope of our work. We found this presentation style to be the most effective solution for our findings and their context to the reader.

That said, we remain open to any additional suggestions for improvement should the reviewer or editor consider it necessary.

2. For all figures: 1) The figure title needs to be below the figure with the figure legend, not above the figure. 2) Instead of using scales of color (e.g., 2e-g), exact amounts need to be labelled. Each wave on these graphs needs to be labelled with a specific amplitude, period, or whatever is being described. This also applies to every figure except 1. 3) All graphs within figures need to be bigger. Some details are not clear within the images.

We thank the reviewer for the suggestions regarding figure presentation. We have carefully addressed all points as follows:

Figure Titles: *We have moved all figure titles below the figures and included them as part of the figure legends, as requested.*

Exact Amounts and Labels: *For all relevant figures (e.g., Figures 2e-g and others), we replaced color scales with **exact values** (e.g., amplitudes, periods, or other parameters) to ensure clarity. Each wave or data series is now explicitly labeled with the specific values being described.*

Graph Size: *We have increased the size of all graphs within the figures to improve readability and ensure that all details are clear and accessible.*

We appreciate the reviewer's careful attention to detail.

3. Figure 1 is mislabelled. 1) There is no reference to panel C. 2) Left and right are not clear in panel A, as there are several sections within this panel.

We thank the reviewer for identifying the mislabeling in Figure 1. This has been corrected in the updated version of the manuscript.

4. Figure 2: 1) What is the purpose of the dial in 2a? 2) What is the Y axis in 2d? “Circadian Clock” is not a descriptive unit. The same goes for several of the graphs, including “ToD Response” and “ToD MR.” Y axes should have accompanying units where applicable.

We thank the reviewer for the opportunity to clarify and improve our figures.

1. *Figure 2a: We acknowledge the confusion caused by the dial in the sketch from Figure 2a. To avoid any ambiguity, this element has now been removed.*
2. *Y-Axis Labels in Figure 2d and Related Graphs:*
 - a. *The y-axis label in Figure 2d has been updated from "Circadian Clock" to the more descriptive term "Circadian Signal."*
 - b. *Across all relevant figures, we have revised the y-axis label "ToD Response" to "Normalized ToD Response". This represents the ratio of cell counts at the final time point to the initial time point, which makes it unitless.*
 - c. *To ensure consistency, we normalized the ToD response to its initial value, enabling comparable scaling across different experimental conditions.*
 - d. *Consequently, the ToD MR (Maximum Range) is also unitless, as it reflects the difference between the maximum and minimum ToD responses.*

These changes have been incorporated into both the updated figures and the main text (see lines 657–659).

Rationale for omitting units in schematic representations:

We decided to keep the axes unit-free in these sketches to emphasize their conceptual and qualitative nature, focusing on trends and relationships rather than precise measurements. We think that including units like "a.u." could imply unintended quantification and cause confusion about the illustrative purpose of these figures.

We hope that these revisions address the reviewer’s concerns and improve the clarity of the figures. However, we remain open to further suggestions if the editor or reviewer feels additional adjustments are necessary.

5. Figure 4: 1) What is the purpose of infinity hours? 2) It is unclear why there are references to two different times following drug preparation. What is the difference between the different curves being prepared at different times and the time labelled on the X axis?

We have addressed all suggestions as follows:

Figure 4 – “Infinity Hours”:

We have clarified the purpose of the term "infinity hours". This originally referred to the case where the drug does not lose its effect over time, which corresponds to an extremely slow decay rate or effectively a stable drug condition. To improve clarity, we replaced “infinity hours” with "no decay", which we believe is a more precise description.

Clarification of Time References:

The references to two different times following drug preparation were indeed unclear. We have revised the figure legend and main text to explicitly explain the difference:

- *The X-axis time in Fig. 4a, c, and h represents the duration over which the specific effect is measured (e.g., experimental observations or simulations).*
- *The term "Preparation age" refers to the time elapsed between drug preparation and its application. This parameter is critical for illustrating how drug stability can affect the observed results, as drugs with faster decay lose effectiveness over time, while more stable drugs retain their effect for longer durations.*

6. Figure 5: What is the difference between the solid and dashed curves in 5e?

We apologize for the lack of clarity in the previous version. The dashed line represents the ToD response relative to the initial time point, while the continuous line corresponds to the ToD response relative to the treatment time. This information has now been incorporated into the labels of Fig. 5e.

Overall:

Major Comments:

1. The multidisciplinary nature of the research (i.e., the combination of computational biology, pharmaceutical sciences, cancer biology, and circadian biology) is excellent, however, the language used in the paper appears to be more consistent with a mathematical perspective rather than a life-sciences perspective. As a result, the actual meaning of the statements made in the paper is quite ambiguous and allows for multiple competing interpretations of what is being said, depending on from which linguistic paradigm you approach the paper.

We appreciate the reviewer's comment regarding the multidisciplinary nature of the manuscript and the potential ambiguity caused by mathematical phrasing. We have carefully revised the text throughout the manuscript to adopt clearer and more consistent wording that bridges the gap between computational, pharmaceutical, and biological disciplines. Specifically, we:

- *Reworded mathematical terms to emphasize their biological relevance, e.g., replacing "steepness of survival curves" with "enhanced cancer cell killing rate", and similarly with several parameters of our model.*
- *Provided explicit definitions for key terms to minimize ambiguity and ensure accessibility to readers from diverse fields, e.g. preparation age and decay rate are now clearly stated.*

We hope that these changes help to ensure that the meaning and context of each statement aligns better with life sciences terminology, without compromising the accuracy of the computational approach.

2. Furthermore, there are many instances in which the impact of experimental conditions is described as a function of graphical transformations, rather than the relevant biological principle represented by said graphical transformation. This can be remedied by clearly stating the practical and relevant points being demonstrated graphically, such as replacing “greater steepness in the survival curves” (line 180) with “enhanced cancer cell killing rate”. While the authors might consider graphical transformation the most intuitive means of understanding the data, the indirect approach to describing data is more confusing than anything else to a researcher not familiar with mathematical jargon. Considering the journal in which this paper is to be published, a greater degree of linguistic assimilation would be appreciated.

We have revised the manuscript to ensure that our results are more directly connected to their biological implications, making the presentation clearer and more accessible to a broader audience. For example:

- *Terms such as the steepness as directly suggested by the reviewer but also “circadian signal amplitude” have been explicitly described as the strength of circadian regulation in biological processes, emphasizing their functional role in modulating drug response.*
- *In all relevant sections, we clarified how graphical outputs, such as changes in slopes or amplitudes, relate to specific biological phenomena (e.g., drug sensitivity, cell killing rates, or stability over time).*

We hope these revisions improve the readability of the manuscript for life scientists while retaining accuracy for computational audiences. However, we remain open to further adapting the language if the editor or reviewer believes additional improvements are necessary.

3. There appears to be some patently incorrect definitions in the paper. For example, the Hill Coefficient (introduced in line 179) is, in this paper, defined as a “survival sensitivity slope” and is said to represent “the sensitivity of a drug’s effect to its concentration” while scientific literature defines the Hill Coefficient as “a dimensionless parameter that has long been used as a **measure of the extent of cooperativity**” (DOI: 10.1016/S0076-6879(08)04207-9). It is not entirely clear what the authors mean when they use the term “Hill coefficient” considering this apparent misuse of the term. Additionally, the use of “strength” to describe the bioavailability/concentration/dose of a drug is another example of imprecise language. Strength in the context of pharmaceutical sciences refers to “the amount of drug in a given dosage form” such as in the description of “extra-strength (higher dose) Tylenol”. It is possible that some of these errors have arisen as a result of correct terms being “lost in translation” between German and English, however, more attention should be given to the context-dependent definitions of the terms used to improve the quality of this paper.

We thank the reviewer for carefully pointing out these inaccuracies and for helping us improve the clarity and consistency of our manuscript. We have now corrected all occurrences where definitions were imprecise or potentially misleading:

- *The Hill coefficient is now correctly defined as a “dimensionless parameter used to describe the extent of cooperativity in dose-response relationships.”*

- *We have removed most appearances of the term “strength” when referring to dose or concentration, replacing it with more precise terminology such as “dose” or “drug concentration” to align with standard pharmaceutical definitions and avoid misinterpretation.*

We are grateful to the reviewer for helping us refine our language. These changes significantly clarify our message and improve the manuscript's readability for a broader, multidisciplinary audience. We remain open to any additional suggestions from the editor or reviewer to further enhance the presentation of our work.

Reviewer #2 (Remarks to the Author):

Gutu and colleagues aim to better understand circadian modulation of drug sensitivity. The authors combine modeling with a re-analysis of a data set they recently published in Nature Communications.

However, when I tried to focus on what was explicitly done/learned in this paper (as opposed to the already published data), it seemed as if the primary methods were to assume a sinusoidal form for various parameters (one at a time) and plot the results of those sinusoidal assumptions. Similarly, some of the other conclusions seemed well known. The language of the paper makes it seem that they did more extensive “simulations” than this and given the author’s well-established expertise I worry that I may have misunderstood. I have tried to highlight points where clarification may help.

We appreciate the reviewer's insightful feedback and have revised several sections to clarify and emphasize the key contributions of our study. In the revised version, we now explicitly state what aspects of the work are novel contributions and ensure that our methods and findings are properly contextualized against the existing literature. We will also adapt the language to more accurately describe the scope and methodology of our simulations, ensuring clarity and avoiding any potential misunderstandings. See changes in the Introduction line 78-94, in the Discussion lines 379-389.

These updates make the scope and originality of our work more evident. Please let us know if further clarification is needed.

General:

1. Despite the aim to model temporal complexity, the models appear to assume fixed and unchanging drug effects based on the age of the drug and the circadian state of the cells at the moment of drug administration. (Indeed, the analytic solution for $x(t)$ in line 495 is only valid if these parameters are constant over the course of the simulation). For example: the simulated growth curves in 2b (and elsewhere) seem to reflect an assumptions of a constant exponential growth (with different but constant rates depending on the time of initial administration). But the experimental system is closed with drug remaining in the dish over time. The authors exposition would seem to suggest a more complicated model with different effective growth/local killing rates at different circadian times as the simulation advances in time and some fraction of the drug remains in the dish? But I do not think this is what was done?

We apologize that the language in the original version of our manuscript did not clearly convey the assumptions underpinning our modeling approach. You are correct in observing that our models assume a fixed and unchanging effect of drugs based on the age of the drug and the circadian state of the cells at the time of administration. This simplification was intentionally chosen to isolate and focus on the current set of temporal components, at the time of treatment, influencing time-of-day responses. However, we acknowledge that this approach may not encompass the more complex and dynamic interactions that occur in biological systems.

In other words, given the multiple temporal components that can affect the time-of-day responses, we opted for a simplified model where these parameters remain constant. This decision was made to facilitate a clearer understanding and initial exploration of the primary

mechanisms at play. Nonetheless, we agree that more realistic and complex scenarios, which include varying drug effectiveness and cellular responses over time, need to be incorporated in future studies to fully capture the dynamics of drug-cell interactions.

We have revised several sections of our manuscript to stress this point, clearly articulate the assumptions of our modeling approach and address the limitations of our approach. See changes in the Discussion lines 369-378.

2. Most of the experimental correlations shown in the main figures (correlations between circadian parameters and various drug effects (e.g 2m, 2n) seem to be primarily driven by 1-2 points with very high leverage. Moreover, each point is a completely different cell type with many differences in biology beyond the amplitude of a single circadian luciferase reporter. Thus, it is hard to be confident that the differences seen in those 1-2 more extreme cell types are driven by the circadian differences between the cells.

Thank you for your observations regarding the experimental correlations presented in figures such as Fig. 2m and 2n. We acknowledge that these correlations may be influenced by a small number of data points with high leverage and the biological diversity among the cell types, each with unique characteristics beyond the amplitude of a circadian luciferase reporter. Additionally, we note that the parameter distribution is not uniform, which could also influence the observed trends, and apply a different method to compute correlation (as recommended by the reviewer in point number 6 below).

Our analysis was designed as a targeted investigation where the assumptions of our model might best capture the dominant effects driving the observed experimental ToD responses. This approach allows us to highlight specific contexts where our simplified model aligns with experimental outcomes, while recognizing that biological systems are inherently more complex. Future studies, particularly those utilizing isogenic cell lines with tunable circadian parameters, will be essential for further validating and refining these insights. Such studies would provide a clearer understanding of how circadian features influence time-of-day drug response curves under more controlled conditions.

Specific Points:

(1) The equation in solution for $x(t)$ in line 495 is only valid if the parameters in line 489 are constant. But this is not the case (as I understand it) for a system with circadian variation in activity over the course of the trial. I thought this is what you wanted to model? Please clarify – and highlight in the text if the growth rates/kinetic/drug efficacy terms are held fixed for the duration of each simulation

We regret not having clearly stated our model assumptions in the previous version of our manuscript. Our model assumes that the growth rates, kinetics, and drug efficacy terms remain constant over the duration of each simulation. This simplification was intentionally chosen to isolate the effects of drug age and circadian state at the time of administration, focusing on these key temporal components influencing time-of-day responses. We acknowledge that this does not capture the full complexity of systems with circadian variation over time, and we highlight in the text that incorporating such dynamic interactions will be an

important step for future studies. We have now clarified this approach in our manuscript, see lines 152-155.

(2) In describing the equation on line 500 you write that you are trying to model a sinusoid modulation of the ic_{50} . But the equation says $c(t)$ is equal to the terms on the right-hand side? What exactly is $c(t)$ in - for example - the equations in 505 and 505?

We thank the reviewer for pointing this out. In response, we have now clearly defined the modulation of IC_{50} and clarified the meaning of $c(t)$ in the context of the equations. Additionally, we have rewritten the equation derivation step by step to provide a clearer and more transparent explanation of how the terms are derived. We hope this revised presentation resolves any ambiguity and improves the overall clarity of the section.

(3) In Panel 2F and the related description/discussion you conclude that prolonging the period doesn't affect the ToD response until the period is a little beyond 30 hours. It looks like you simply assumed that the influence at time 0 is fixed – and then only compute the influence of TOD for a 24 hour period. Thus if the peak efficacy occurred after the 24 hour period you evaluated, it would not show up in the way that you assess? Is this correct? If however, you did search over the course of the cells full period - there would be no effect?

We appreciate this opportunity to clarify certain aspects of our analysis and are grateful for your insightful observations. In our study, we modeled cell growth under a cytotoxic drug with fixed circadian modulation, evaluating the drug response within a standard 24-hour cycle. This approach was used even when the circadian modulation extended beyond 24 hours, to ensure clarity and interpretability in assessing time-of-day (ToD) effects.

You are absolutely right in suggesting that analyzing the ToD response over a time window that matches the actual period of circadian modulation could provide different insights, particularly as models deviate from the standard 24-hour rhythm. Our method reflects common experimental setups where the intrinsic period of circadian modulation in the samples isn't always considered. This oversight can significantly impact the shape and interpretation of ToD curves based on a 24-hour framework. In response to your feedback, we are now updating the main text of our study to clarify these points more explicitly.

We have now clarified this important point in the Discussion section of our manuscript line 454-465.

We hope our study lays the groundwork for future investigations into how deviations in rhythm length might influence ToD responses. Thank you once again for your valuable feedback, which not only enriches our discussion but also helps steer future research directions in this field.

(4) It is hard to characterize a tumor state at a given time (e.g tumor state every midnight) if the tumor's entrained period is not 24 hours. Each passing day the tumor's clock would shift relative to our wristwatches. If we do assume that tumor clocks might not have an entrained 24 hour period – then the benefits/drawbacks for timing personalized therapy (as the authors emphasize in their discussion) would likely be measured relative to the longer time period?

(and given the authors model would not affect TOD) Is this all correct?. In your in-vitro case if I waited for the second day to do the analysis - I would assume I would get different answers as the relative period has shifted.

Thank you for your insightful comments regarding the characterization of tumor states and the timing of personalized therapy in relation to unstable, non-24-hour circadian rhythms. You have accurately highlighted the complexity involved in characterizing tumor states at consistent times each day when the tumor's intrinsic circadian period may not synchronize with a 24-hour cycle. This misalignment means that the tumor's biological clock could shift each day relative to conventional timekeeping, which could complicate the timing for personalized therapy.

Our current model does not account for the variability in the tumor's circadian rhythm over multiple days. This oversight means that ToD responses could vary if measurements are taken on different days. Furthermore, the repetitive administration of treatment doses might induce phase shifts in tumor cells, necessitating ongoing adjustments and characterization of the tumor cells' phase response curve.

These complexities underscore the need for more comprehensive integrative models that can accommodate such dynamic scenarios. Such scenarios are beyond the scope of this current work, but your feedback is instrumental in guiding future enhancements to our modeling approach.

Recognizing these very important challenges, we are updating our manuscript to include a new section in the discussion that addresses these points explicitly. This addition will help clarify the limitations of our current model and the implications for clinical practice. See 466-483

(5) A few plots compare estimated circadian features based on dynamic luminescence with drug effect (2m, 2n, 2o,). Separate estimates based on Per2-luc and BMAL-luc transfections are given in the tables. Which estimates were used in the plots? Why was that one chosen? Was it an average? A quick look at the data suggests that the PER and BMAL amplitudes are poorly correlated. It is not clear why the authors assume that the amplitude of a single circadian reporter reflects the amplitude of the clocks influence on drug effect.

We apologize for our lack of clarity in the original manuscript. In this work we first examined all the correlation coefficients between parameters extracted from the Per2 and BMAL1-luciferase signals independently, and the Time-of-Day responses for different drugs. We then selected those cases where these correlations were strongest. Specifically, for Fig. 2m, we used BMAL1 estimates; for Fig. 2n, Per2 estimates; and for Fig. 2o, BMAL1 estimates. We now clarify this better in the corresponding captions of these figures.

Thanks for the opportunity to clarify our approach regarding the use of a single circadian reporter's amplitude to study the circadian clock's influence on drug effects. In our study, we explore the extent to which the amplitude of an individual reporter can be used as an indicator of this influence across different drugs and cell lines. While we are aware that this method represents an oversimplification, circadian reporters are well-established proxies for the overall activity of the circadian system. The amplitude of these reporters has been shown to correlate with the expression patterns of clock-controlled genes¹⁻⁴, which are crucial for

various cellular functions, including drug metabolism⁵. By measuring the amplitude of a circadian reporter, we aim to provide an insight into the general strength and impact of the circadian clock on pathways critical to drug efficacy.

In other words, while we acknowledge that the relationship between reporter amplitude and downstream clock-controlled pathways may not be strictly linear or identical for all pathways, it provides an approximation for the clock's systemic impact on drug effects.

We have now included this discussion in the revised manuscript (lines 174-178).

(6) Concretely: Given the apparent dependence of the various correlations on a few outliers, I would ask that rank based (Spearman) correlations and significance measures be reported instead of Pearson correlations for all these plots (and the plots 3h and i). Relatedly the amplitude of the luciferase signal is (I would assume) strongly dependent on the efficiency of the viral transfection. Was there some method to account for this confounder in the analysis? Please specify? The authors should add to the discussion as to why the amplitude of the given circadian reporter is expected to correlate with the circadian amplitude of the particular drug's effect.

Addressing Concerns Over Pearson Correlation Coefficients

We thank the reviewer for pointing out the potential influence of outliers on the Pearson correlation coefficients used in our analysis. In response, we have computed Spearman correlation coefficients, which are less sensitive to outliers, along with their significance levels. These results have been integrated into the updated version of our figures. For comparative purposes, we have retained the original Pearson correlation coefficients in the figure captions, allowing for a comprehensive view of both statistical approaches.

Efficiency of Viral Transfection and Its Impact on Amplitude Estimations

We acknowledge the critical importance of the efficiency and variability in the generation of circadian reporter cell lines using lentiviral transfection, as these factors can significantly influence the amplitude of the luciferase signal. To address these concerns, we have implemented several robust measures:

- ***Efficiency of Transfection Protocols:*** *Collaborating with the Kramer Lab at the Charité – Laboratory of Chronobiology, we have utilized their expertise and standardized protocols to ensure high transfection efficiencies and reduced variability in reporter expression.*
- ***Cross-Validation of Cell Lines:*** *We have corroborated our circadian measurements by comparing them with results from similar cell lines independently developed in other laboratories, such as MCF10A, MCF7, and U-2 OS WT and KO. This cross-validation ensures that our cell lines display consistent circadian patterns, aligning with established findings.*
- ***Repetition and Consistency:*** *Originally documented in Ector et al. (2024)⁶, we addressed concerns about variability by repeating the transfection process for the Bmal1-Luc reporter in a set of wild-type cells. These repetitions confirmed that our protocol provides consistent luciferase signals across trials.*

- **Population-Level Analysis:** Our analysis of luciferase signals is conducted at the population level using non-clonal cell lines. This approach effectively minimizes the impact of any individual adventitious insertion, utilizing the averaging effect across the population to dampen outliers and reduce signal variability.

These strategies significantly mitigate potential issues related to transfection efficiency and random viral insertion, providing a reliable foundation for interpreting the circadian amplitude measured in our assays.

Amplitude Relationship with Drug Responses

Central to our study is the assumption that circadian regulation affects the effective concentration of a drug, leading to an expected correlation between the amplitude of our circadian reporter and the Time of Day (ToD) curve amplitude. This assumption serves as a guiding principle for our analysis, enabling us to explore the extent of this relationship across various drugs and cell lines.

Although this model may be simplistic, it offers a structured framework to systematically examine where our predictions coincide with experimental outcomes. This approach helps us identify which drugs are most affected by circadian rhythms, enhancing our understanding of circadian influences on drug efficacy.

We hope this explanation clarifies the rationale behind our methodology and aids in understanding the extent of the relationship between our circadian reporter amplitude and drug response amplitudes in practical applications.

Addressing these concerns, we have revised our several sections. See updated Figs. 2 and 3 and also lines 409-416

(7) The major outcome of the analysis of drug properties (Figure 3) Is (I think) simply that if a drug is sufficiently efficacious – there will be less of a TOD difference if it is administered at either its peak or trough circadian efficacy. In the extreme example – if the drug kills all the cells regardless of administration time – there would be no TOD dependence. Is this the point? Please clarify. If this simple (well-known) fact is what is being highlighted – it is likely obscured by plotting only the TOD variation and not the killing rates at the two most extreme times. Consider adding those plots.

We apologize for any confusion and appreciate the reviewer's feedback, which highlights the need for clearer explanations in our presentation of Figure 3. This figure is designed to investigate various independent parameters affecting drug responses, such as reference drug concentration, drug sensitivity slope (also known as the Hill coefficient) and, maximal drug effect (E_{max}). These parameters can be viewed in different scenarios as indicators of drug efficacy.

We analyzed each parameter impact on the ToD curves which results in very different behaviors. For example, our model reveals that for the reference drug concentration, higher doses do not necessarily correlate with increased ToD curve amplitude. Instead, there is an optimal concentration that maximizes the ToD curve amplitude. Additionally, we observed that a higher Hill coefficient correlates with a stronger amplitude response.

Regarding the Emax, our observations indicate that sufficiently potent drugs produce a pronounced difference in ToD responses. In contrast, drugs with lower efficacy, as measured by killing rates, show negligible differences in ToD responses. Recognizing that the initial use of Emax (indicating the highest drug concentration) rather than directly referencing killing rates, which are inversely proportional, may have led to some misunderstandings, we have revised the figure captions to include killing rates for better clarity.

This adjustment should aid in accurately conveying the dynamics between drug potency, efficacy, and their circadian responses as captured in our simulations. See updated Figure 3 and lines 398-403.

(8) I am hopeful that there are some unique/novel points in Figure 5. This is also where there is new experimental data. However, it appeared like the authors were showing that if you wait a while before adding drug, the cells will have had time to replicate, and if not accounted for, this will influence the computed efficacy. This would seem to be a trivial result. I worry I missed something. Could you please explain the take-a-way beyond this? Perhaps it can be less concise if needed.

We apologize for not providing a clearer explanation of the message conveyed by Figure 5. To clarify, the model and experimental data highlight two key points: first, that the ToD response is larger directly affected by the division rate of cells, and second, that when the drug response is analyzed without considering the correct referential time, the outcome can be obscured, potentially leading to incorrect conclusions. We emphasize the importance of calculating the drug response based on the treatment time rather than the initial time of the experiment, as using the latter can cause confusion and result in inflated estimates of drug efficacy (lines 428-429 and 433-435).

(9) Some of the discussion was overstated

For example:

“We observed that alterations in circadian rhythm properties affect time-of-day drug sensitivity, resulting in varying ranges of resistance or sensitivity. Specifically, increasing the amplitude of the circadian clock enhanced the maximum range of the time-of-day drug response, in contrast to changes in the period or decay rate of the clock.”

As I understand it, the authors specifically assumed that the core clock amplitude influenced the circadian amplitude of drug responsiveness in their mathematical models. In the data (Torin 2) the correlation the measured clock amplitude and ToD effects was driven 1-2 apparent outliers. I am not sure this was really “observed” so much as assumed in the modeling.

We apologize for any overstatement. We have now revised that statement to provide a more conservative, accurate and balanced presentation of our results (see lines 390-393). Additionally, the new calculation of correlation coefficients using the Spearman rank-based method still demonstrates a strong and statistically significant correlation, even after accounting for outliers. As a result, we continue to observe trends that are consistent with the model outcomes.

"Recent research indicates that circadian medicine offers significant health benefits over standard care ". Recent work has highlighted the potential. To my knowledge, very few (2-3?) clinical studies in specific disease have shown small gender specific benefits. I would be careful not to overstate this.

We apologize for overstating these findings from the literature. We have revised this section and added appropriate citations and more carefully rewritten the statement (see lines 486-487).

"we are the first to highlight the importance of incorporating these factors into experimental protocols"

Several authors have highlighted the importance of drug half-life/stability, pharmacokinetics, and pharmacodynamics. Several in-vitro studies have considered different doses. Many in-vitro and in-vivo therapeutic trials (beyond those in circadian medicine) have worked to characterize the killing effect of a chemotherapeutic based on the number of cells (or tumor mass) when the treatment is administered.

We apologize for the overstatement and thank the reviewer for bringing this to our attention. We have corrected this in the revised version of the manuscript (see lines 484-485).

(10) The limits of the study should also be discussed

We concur with the reviewer's suggestion regarding the need to discuss the limitations of this study. We have now included several points addressing this in the revised manuscript (see lines 384-389 and 445-453 and 466-483).

Reviewer #3

In this manuscript, the authors investigate potential drivers for Time-of-Day drug sensitivity in human cells, i.e., drug efficacy and/or toxicity. Such an investigation falls under the umbrella of chronotherapy, a framework postulating that drug efficacy/toxicity varies according to its administration time, and importantly, to the body's internal circadian rhythms. Indeed, circadian modulation also happens at various levels like cell growth or DNA repair, thus influencing drug effectiveness. Moreover, the different mechanisms involved in how the drugs themselves are processed may be under circadian regulation, impacting drug stability and drug properties. This inherent complexity warrants a careful study to precisely decipher the overall contribution of the circadian clock to drug efficacy. This was achieved by the authors by leveraging a combination of mathematical models and experiments.

1. As for a general impression, the paper is well-written and smoothly organized. I would say that the structure of the paper itself, a sequence of reported findings regarding tedious computational and biological experiments, felt difficult to read at times. But to add some nuance, I agree with the view of the authors on this careful study being necessary. Directly quoting the paper, "previous drug screening approaches often neglect the presence of multiple time-dependent confounding factors, which can severely limit and bias the outcomes. Therefore, it is crucial to investigate how these factors shape the time-dependent dynamics of drug response." and "A comprehensive, transdisciplinary approach that integrates experimental data with mathematical models is necessary to advance this field".

Thank you for your thoughtful review and valuable feedback on our manuscript. We sincerely appreciate your interest in our research and the time you dedicated to providing detailed remarks. In response, we have revised several sections of the manuscript to enhance readability and clarity.

2. Regarding the way the circadian clock is integrated into the models, exogenous rhythms were employed, using damped cosine functions. A decay parameter, phase, amplitude, and mean level parameters govern these oscillations. One can then investigate the effect that different circadian rhythms may have on Time-of-Day drug sensitivity by varying these parameters. Such an approach is an abstraction of endogenous circadian mechanisms, resulting in small-sized models that are easy to fit, but less actionable: if the cosine amplitude appears to be a sensible parameter, how would one experimentally verify that? E.g., which exact member of the circadian clock should be intervened over? This being said, while there exists work attempting to be "more mechanistic" in the way the circadian clock acts on drug sensitivity [1], these often result in large models, highly nonidentifiable, and difficult to fit, which require large amounts of data from many species. Furthermore, it can also be the case that going into such a level of detail is not even required for drawing impactful conclusions.

Thank you for this insightful and thoughtful comment, which precisely captures the balance between the limitations and advantages of our modeling strategy. We are grateful for the reviewer's recognition of the suitability of our approach for the objectives of this study. In response to these valuable points, we have expanded the discussion section to address the critical aspects of our modeling strategy in greater detail. See updated Discussion section lines 379-389.

3. Given all these reasons, I believe the way the authors formulated the study to be best suited. I would recommend:
 - A. Acknowledging the existence of such works.
 - B. Including an explanation as to why exogenous rhythms were considered rather than endogenous rhythms. Currently, an equation describing circadian modulation is included, but the latter does not come with any justification.
 - C. Mentioning the existence of other approaches for representing exogenous rhythms. (Dampened) Cosine waves have been used to capture circadian modulation for a long time, as they provide a simple approach with somewhat interpretable and actionable parameters (see discussion above). Yet, not all oscillations are best represented using cosine waves, e.g., periodic rhythms can be observed with asymmetric rise and slope parts. State-of-the-art modeling tools for that matter now heavily rely on Gaussian Processes [2,3], and employing a different form for oscillations might change the results. To be clear, I am not asking the authors to carry out again all the analyses using a different model. A sentence mentioning that the findings can be influenced by how flexible the model class is for the oscillations would be welcomed, thus providing some nuance to the findings.

Furthermore, we have updated the manuscript to address each of the subsequent comments as follows:

- A. *We now clearly refer and cite the existing literature regarding more mechanistic modeling approaches. See line 449.*
- B. *We thank the reviewer for this insightful comment. We thank the reviewer for this insightful comment. In response, we have clarified and expanded the discussion in the manuscript to address the potential sources of circadian modulation, distinguishing between endogenous and exogenous contributions. Our focus is on modeling the impact of circadian modulation on the growth dynamics of cells, mediated through cytostatic or cytotoxic effects. This approach inherently abstracts the source of the modulation, whether endogenous or exogenous. For example, it could represent an exogenous circadian process affecting the drug concentrations reaching tumor cells over the course of the day, an endogenous circadian clock modulating drug uptake into the cytoplasm or nucleus, or an endogenous circadian regulation of damage response pathways. These scenarios are captured in our model as circadian-modulated effective drug concentrations within the same flexible framework. See lines 115-123.*
- C. *We thank the reviewer for highlighting this important point. We agree that the choice of model for representing circadian oscillations can influence the findings, particularly when the oscillatory patterns deviate from simple cosine waves. In response, we have added a statement in the discussion to acknowledge this limitation and reference alternative approaches, such as Gaussian Processes, which allow for more flexible representations of circadian rhythms. See lines 445-453.*

4. Finally, It is worth mentioning that all the data used throughout the paper was made available by the authors (as Excel sheets). Moreover, the authors created a GitHub repository containing the code used to generate all figures. The code is clean and readable. I did not run every script, but the few I launched worked fine. I would suggest adding a README.md file, indicating:

- What are the necessary packages to successfully run the code
- In what directory should each data xls file be placed.

Thank you for your valuable feedback. We have now added a README.md file in the GitHub repository to include the requested information.

Abovementioned references:

- [1] Hesse et al. A mathematical model of the circadian clock and drug pharmacology to optimize irinotecan administration timing in colorectal cancer. Computational and Structural Biotechnology Journal (2021)
- [2] Sahay et al. An improved rhythmicity analysis method using Gaussian Processes detects cell-density dependent circadian oscillations in stem cells. Bioinformatics (2023)
- [3] Durrande et al. Detecting periodicities with Gaussian processes. PeerJ Computer Science (2016)

References:

1. Littleton, E. S., Childress, M. L., Gosting, M. L., Jackson, A. N. & Kojima, S. Genome-wide correlation analysis to identify amplitude regulators of circadian transcriptome output. Sci Rep 10, 21839 (2020).
2. Doherty, C. J. & Kay, S. A. Circadian Control of Global Gene Expression Patterns. Annu Rev Genet 44, 419–444 (2010).
3. Li, J. Z. et al. Circadian patterns of gene expression in the human brain and disruption in major depressive disorder. Proceedings of the National Academy of Sciences 110, 9950–9955 (2013).
4. Hurley, J. M. et al. Analysis of clock-regulated genes in Neurospora reveals widespread posttranscriptional control of metabolic potential. Proceedings of the National Academy of Sciences 111, 16995–17002 (2014).
5. Fagiani, F. et al. Molecular regulations of circadian rhythm and implications for physiology and diseases. Sig Transduct Target Ther 7, 1–20 (2022).
6. Ector, C. et al. Time-of-day effects of cancer drugs revealed by high-throughput deep phenotyping. Nat Commun 15, 7205 (2024).

REVIEWERS' COMMENTS:

Reviewer #1 (Remarks to the Author):

The authors have done a very good job at addressing all of the concerns that were raised. I'm satisfied with their responses and would recommend it be published.

We sincerely appreciate the reviewer's positive feedback. We are glad that the additional responses and clarifications we provided have addressed their concerns.

Reviewer #2 (Remarks to the Author):

The authors have sufficiently addressed my concerns.

We greatly appreciate the reviewer's positive feedback.

Time Matters: Understanding the Drivers of Time-of-Day Drug Sensitivity in Human Cells

Summary: This paper attempts to characterize the time-dependent cellular and pharmaceutical factors that influence chemotherapeutic drug efficacy in various human cancer cell lines when administered at various timepoints throughout a 24-hour period using both predictive computational modeling and an *in vitro* experimental approach.

Abstract:

Minor Comments:

1. I would challenge the notion that chronotherapy has yet to be fully integrated into clinical practice due to insufficient basic science knowledge concerning why it works. Clinically, the burden of proof needed to implement a therapy is that it is safe and that it actually works, not that the specific mechanism through which it safely works is thoroughly understood. While I appreciate the sentiment that basic biology is valuable in optimizing therapies, it is not a primary challenge that chronotherapy faces insofar as its wide-scale implementation is concerned.

Introduction

Major Comments:

1. Lines 45-46, a citation is needed for each of the circadian modulation mechanisms (immune response, pharmacokinetics, and stress response pathways).
2. Line 80: The specific characteristics of the circadian clock need to be listed.

Results

Major Comments:

1. Line 98: Define reference concentration. This is defined later in the paper but needs to be clearly stated when introduced.
2. Line 101: The authors state that their approach represents how drug effectiveness varies at different times of day. I'm not sure that this can be definitively stated because 1) the authors do not test any drugs within biological organisms and 2) this is in reference to testing difference concentrations, which is extrapolated into efficacy.
3. Any time circadian period is mentioned, it is always way too long. This paper is examining drug efficacy in humans, whose circadian periods can only be 22-26 hours, even in extreme cases.
4. Line 112: What are the ToD values being referred to here?
5. Line 118: The circadian parameters need to be specifically stated.
6. Line 124: 1) Please define the threshold referenced here. 2) Define circadian decay signal.
7. Line 131: The sentences beginning "Here we extracted from each..." is not clear. Consider rewording this to clearly communicate your point.
8. Line 129: Please name the specific circadian clock signals recorded.
9. Lines 178-194: The stated definition of Hill coefficient is wrong. I would suggest re-examining this entire paragraph.
10. I would recommend describing your results in terms of circadian relevance rather than just as a function of graphical transformations. This would make your manuscript more relevant to biologists rather than mathematicians.
11. Lines 206-207: Wouldn't higher decay rates lead to higher survival? This should be stated more clearly.
12. All simulations with drugs do not include which drug was used in the simulation. Drugs are very different and have different properties, and it needs to be indicated which one is being used for these data.

Minor Comments:

1. Lines 110-111: The authors state, "Increasing amplitude of circadian oscillations results in ToD responses with higher amplitudes." This seems inherently intuitive and does not need to be stated.
2. Line 127: What are the predictions?
3. Line 156: The phrase "dose strength" is redundant.

Discussion

Major Comments:

1. Line 331: What does the phrase “stress sensitivity regulation” refer to? Is this what was tested? If so, then how? If not, new concepts relating to the data should not be brought up for the first time in the discussion.
2. Line 369-370: The statement concluding with “...benefits over standard care.” needs a citation.

Methods

Major Comments:

1. The authors need to state how they are measuring circadian rhythm parameters as well as define these parameters.

Minor Comments:

1. Line 418: There is some sort of issue with formatting the concentration of dexamethasone. It seems to be 1M dexamethasone, but please clear this up.
2. Line 503: What is the opposite of 1? Try using a different word like “otherwise” to make this clearer.
3. Line 508: “Concentration strength” is redundant.
4. It might help to show the derivation of the equations used.

Figures

Major Comments:

1. Consider illustrating your computational model alongside experimental results so as to facilitate a greater understanding and appreciation of what it is exactly you are predicting.
2. For all figures: 1) The figure title needs to be below the figure with the figure legend, not above the figure. 2) Instead of using scales of color (e.g., 2e-g), exact amounts need to be labeled. Each wave on these graphs need to be labeled with a specific amplitude, period, or whatever is being described. This also applies to every figure except 1. 3) All graphs within figures need to be bigger. Some details are not clear within the images.
3. Figure 1 is mislabeled. 1) There is no reference to panel C. 2) Left and right are not clear in panel A, as there are several sections within this panel.
4. Figure 2: 1) What is the purpose of the dial in 2a? 2) What is the Y axis in 2d? “Circadian Clock” is not a descriptive unit. The same goes for several of the graphs, including “ToD Response” and “ToD MR.” Y axes should have accompanying units where applicable.
5. Figure 4: 1) What is the purpose of infinity hours? 2) It is unclear why there are references to two different times following drug preparation. What is the difference between the different curves being prepared at different times and the time labeled on the X axis?
6. Figure 5: What is the difference between the solid and dashed curves in 5e?

Overall:

Major Comments:

1. The multidisciplinary nature of the research (i.e., the combination of computational biology, pharmaceutical sciences, cancer biology, and circadian biology) is excellent, however, the language used in the paper appears to be more consistent with a mathematical perspective rather than a life-sciences perspective. As a result, the actual meaning of the statements made in the paper is quite ambiguous and allows for multiple competing interpretations of what is being said, depending on from which linguistic paradigm you approach the paper.
2. Furthermore, there are many instances in which the impact of experimental conditions are described as a function of graphical transformations, rather than the relevant biological principle represented by said graphical transformation. This can be remedied by clearly stating the practical and relevant points being demonstrated graphically, such as replacing “greater steepness in the survival curves” (line 180) with “enhanced cancer cell killing rate”. While the authors might consider graphical transformation the most intuitive means of understanding the data, the indirect approach to describing data is more confusing than anything else to a researcher not familiar with mathematical jargon.

Considering the journal in which this paper is to be published, a greater degree of linguistic assimilation would be appreciated.

3. There appears to be some patently incorrect definitions in the paper. For example, the Hill Coefficient (introduced in line 179) is, in this paper, defined as a “survival sensitivity slope” and is said to represent “the sensitivity of a drug’s effect to its concentration” while scientific literature defines the Hill Coefficient as “a dimensionless parameter that has long been used as a **measure of the extent of cooperativity**” (DOI: 10.1016/S0076-6879(08)04207-9). It is not entirely clear what the authors mean when they use the term “Hill coefficient” considering this apparent misuse of the term. Additionally, the use of “strength” to describe the bioavailability/concentration/dose of a drug is another example of imprecise language. Strength in the context of pharmaceutical sciences refers to “the amount of drug in a given dosage form” such as in the description of “extra-strength (higher dose) Tylenol”. It is possible that some of these errors have arisen as a result of correct terms being “lost in translation” between German and English, however, more attention should be given to the context-dependent definitions of the terms used to improve the quality of this paper.